# Health effects associated with exposure of children to physical violence, psychological violence and neglect: a Burden of Proof study

The health toll of child maltreatment or violence against children (VAC) has not yet been comprehensively evaluated. Here, in our systematic review and meta-analyses, we focused on the health impacts of physical violence, psychological violence and neglect during childhood. Utilizing the Burden of Proof methodology, which generates conservative measures of association while accounting for heterogeneity between input studies, we evaluated 35 associations between VAC and adverse health outcomes, identifying 27 statistically significant links. The associations between physical violence and major depressive disorder, ischaemic heart disease, alcohol use disorder, eating disorders and drug use disorders were rated as moderately weak, reflecting a small effect size and/or inconsistent evidence. The minimum increased risk ranged from 16% for depression to 2% for drug use disorders. Psychological violence showed similar moderately weak associations with drug use disorders (8% minimum risk increase), migraine (7%) and gynaecological diseases (2%). Neglect was linked to at least a 15% increased risk for anxiety disorders. The other 18 associations were weaker due to smaller effect sizes and/or less consistent evidence. Despite the limitations of the existing evidence, our analysis highlights substantial health impacts for VAC survivors, underscoring the need for health system prioritization and continued efforts to eliminate all forms of VAC.

The safeguarding of children against all forms of violence is a fundamental right enshrined in the Convention on the Rights of the Child, alongside other international human rights treaties. Yet, exposure to violence continues to be a harsh reality for children across the world, transcending economic and social status, culture, religion and ethnicity[1]. Worldwide, nearly 1 billion children aged 2–17 years are subjected to physical, sexual or emotional violence or neglect each year[2], with three in four young children experiencing physical punishment or psychological aggression on a regular basis[3]. The most dangerous spaces for children are too often those that are supposed to be safe environments, most notably the home, where parents, relatives and those who are supposed to provide care are frequently the perpetrators[4].

Experiencing and witnessing violence, especially at an early age, can have profound and far-reaching consequences on immediate and long-term well-being. Beyond deaths[5], violence against children (VAC), represented by forms of direct abuse or neglect of children, has been linked to a spectrum of adverse health outcomes, including physical injuries, developmental issues, chronic diseases, childhood pregnancy and a range of mental health conditions[6–13]. Early-life exposure to violence also increases the likelihood of high-risk coping behaviours[14,15], certain unsafe sexual practices[16] and substance use[17], which exacerbate negative health outcomes associated with experiencing violence as a child. The societal impact of VAC is equally substantial, undermining educational attainment and diminishing economic potential[18,19]. Compounding this, exposure to VAC is associated with subsequent

✉e-mail: lsflor@uw.edu

## Table 1 | Summary and policy implications

| | |
|---|---|
| Background | VAC is a critical human rights and public health issue, with nearly one billion children experiencing physical, sexual or psychological abuse or neglect annually. VAC, also referred to as child maltreatment, leads to severe health, developmental and psychosocial consequences, fostering cycles of violence that impact future generations. Yet, the full scale of the crisis is probably underestimated due to challenges quantifying exposure to violence and societal normalization of some abusive behaviours. Here, we systematically reviewed the published literature across seven databases and conducted meta-analyses to assess the health impacts of childhood physical violence, psychological violence and neglect on a diverse range of health outcomes. |
| Main findings and limitations | We quantified the associations of childhood physical violence, psychological violence and neglect with 15, 11 and 9 specific health outcomes, respectively. Out of the 35 risk–outcome pairs evaluated, our results suggest statistically significant increased risk for 11 health outcomes in individuals who experienced physical violence as a child, 10 in those subjected to childhood psychological violence and 6 in those who were neglected during childhood.<br><br>Using conservative Burden of Proof metrics that account for between-study heterogeneity and quantify both the effect size for the association and the strength of the underlying evidence, estimated associations ranged from moderately weak (two stars) for 9 risk–outcome pairs to weak (one star) for 18 pairs. Our conservative interpretation of the data suggests that exposure to physical violence increases the risk of major depressive disorder and ischaemic heart disease by at least 16% and 15%, respectively. Psychological violence exposure increases the risk of drug use disorders by at least 8% and the risk of migraine by at least 7%, while childhood neglect increases anxiety disorders risk by a minimum of 15%. One-star risk–outcome pairs primarily reflect a lack of available data coupled with considerable heterogeneity in the data that are available, suggesting more research is needed.<br><br>Our study was limited by several factors, including variability in how violence exposure was defined across the included studies, which contributes to between-study heterogeneity. Although some discrepancies could be addressed through bias covariates, others may have also affected our results. In addition, treating physical violence, psychological violence and neglect as separate, dichotomous risk factors may oversimplify the complexities of these exposures and their associated health risks, which can be interrelated and vary with severity or frequency. Our analysis was limited to health outcomes that align with the Global Burden of Disease definitions and were supported by at least three published studies. Potential health risks from other forms of VAC or outcomes not covered by the present meta-analysis may represent further unaddressed aspects of the total health burden of exposure to VAC. |
| Policy implications | In light of the profound health impacts of childhood physical violence, psychological violence and neglect revealed by our systematic review, it is critical for policymakers, health professionals and advocates to urgently prioritize the elimination of all forms of VAC. The evidence calls for a comprehensive approach that synthesizes public health policies, socioeconomic strategies and educational initiatives to prevent exposure to violence and to support those who have survived it. With the aspiration to meet the Sustainable Development Goal of eradicating all forms of VAC by 2030, prevention and response measures must be inclusive, ensuring that no child is left behind. This pursuit is not just a moral obligation; it is also a prudent investment towards securing healthier, violence-free futures for upcoming generations. To propel this mission forward, enhancing data collection on the various dimensions of violence and its consequences is imperative to provide the foundation for a global health agenda that prioritizes children's welfare. By committing to adversity-free childhoods and implementing evidence-based interventions that preempt both childhood abuse and its potential continuation into adulthood, we can dismantle the cycle of violence and establish a legacy of health and prosperity that benefits all individuals and societies. |

violent behaviour and violence victimization, suggesting that VAC not only harms the individual victims throughout their life course but also propagates a cycle of violence across generations[20,21].

Although existing estimates of prevalence suggest that the magnitude of the problem is massive, they are probably underestimates of the true extent of VAC, owing to the inherent challenges in accurately assessing the full spectrum of VAC and the likelihood of underreporting due to stigma and fear of reprisal[22,23]. Further, some forms of VAC (for example, corporal punishment) are socially accepted, tacitly condoned or not perceived as being abusive in certain settings, which both normalizes what is a public health risk and fuels a dangerous misconception that VAC remains a marginal and rare phenomenon.

The burden of disease associated with VAC has also been underestimated so far. The 2021 iteration of the Global Burden of Diseases, Injuries, and Risk Factors Study (GBD) estimates that childhood sexual abuse (CSA) is responsible for 2.8 million disability-adjusted life years (a measure of both premature death and years lived in ill health) globally[24]. However, this estimate is based on the association of CSA with a limited number of health outcomes and, hence, underestimates disability-adjusted life years. Further, GBD does not yet account for the loss of health resulting from other types of VAC beyond CSA. These limitations extend beyond the GBD. Most systematic reviews concentrate on the health impacts of adverse childhood experiences as an umbrella term, failing to disentangle the health consequences associated with specific forms of VAC[25,26]. Moreover, these reviews are frequently restricted in the range of health outcomes investigated, with many focusing primarily on reproductive outcomes among women or mental and substance use disorders[26–28]. Therefore, it is crucial to systematically investigate the relationships between the various forms

of violence to which children are exposed and a wide range of adverse health outcomes, allowing a more accurate estimation of the disease burden attributable to VAC. Such efforts are essential in amplifying the urgency for global attention and positioning VAC appropriately in policy and budgeting agendas.

In this study, drawing from a systematic review conducted as part of an initiative by the Lancet Commission on Gender-Based Violence and Maltreatment of Young People to quantify the health burden of violence across the lifespan, we estimated the associations between childhood physical violence, psychological violence and neglect and a diverse set of health outcomes. The health risks associated with CSA are reported in a prior publication[10].

For this purpose, we used the Burden of Proof methodology developed by Zheng and colleagues[29], which yields a conservative estimate of the harmful effects of exposure to VAC, rigorously quantifies the consistency of the underlying evidence and translates the findings into an easily interpretable star rating system, where one star denotes a weak association and/or inconsistent evidence and five stars indicate a strong association supported by consistent evidence. This analytical framework incorporates—among other systematic modelling components—covariate selection and adjustment to account for known variation in input study characteristics and further quantifies remaining 'unexplained' between-study heterogeneity (that is, gamma and its uncertainty) using random effects modelling. Therefore, this method more fully accounts for heterogeneity compared with traditional meta-analytic approaches, providing uniquely robust, credible and well-specified evidence for policymakers and public health professionals dedicated to reducing VAC and its impacts. The main findings and policy implications of this work are summarized in Table 1.

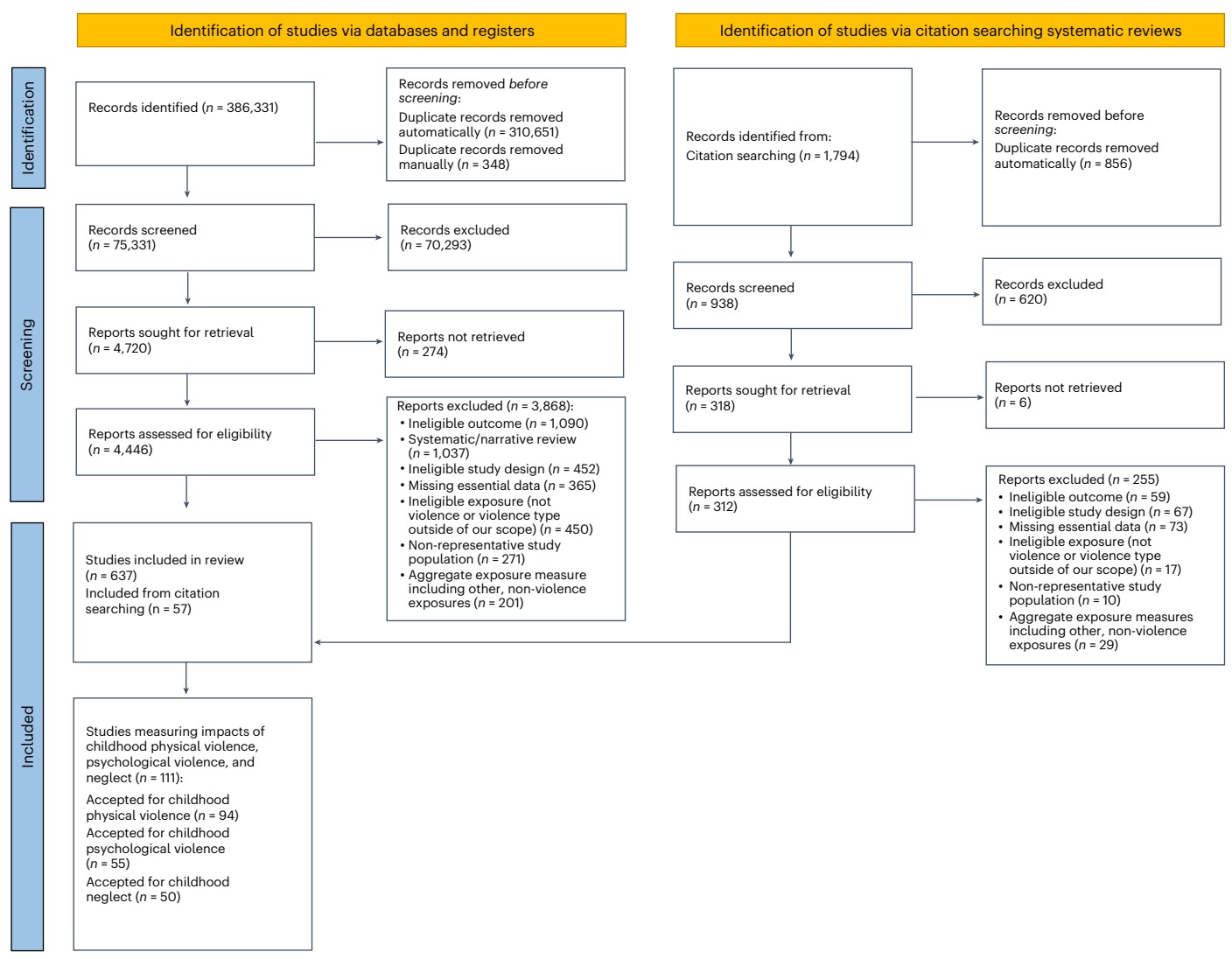

**Fig. 1 | PRISMA flow diagram reflecting VAC studies included in the systematic review on health effects of GBV and VAC.** The PRISMA flow diagram lays out the systematic review process that aimed to identify all relevant literature indexed in seven databases from 1970 through January 2024 related to the health effects of GBV and VAC. The present analysis narrows in on the subset of data identified that pertained to the effects of physical violence, psychological violence and neglect against children on health outcomes with enough data to analyse. These specific counts are reported in the bottom-most box. The diagram was prepared in accordance with PRISMA 2020 guidance from ref. 186. For more information, see http://www.prisma-statement.org.

## Results

A total of 75,331 records were screened as part of an initial systematic review of the literature, which included articles in any language indexed across seven databases over the 54 years between 1 January 1970 and 31 January 2024. Of these records, 638 studies that reported on the association between exposure to any type of violence during childhood or gender-based violence (GBV) and specific health-related outcomes met the predefined eligibility criteria (Methods). All other studies were found to be ineligible, including 452 with an ineligible study design (Fig. 1). While the initial review encompassed literature reporting on any form of violence, the current analysis was narrowed to focus on the health effects of physical violence, psychological violence or neglect occurring before the age of 18, regardless of the perpetrator of the abuse. In alignment with the International Classification of Violence against Children (ICVAC)[30], established by the United Nations Children's Fund (UNICEF), these forms of violence respectively refer to deliberate, unwanted and non-essential acts of physical violence; verbal or non-verbal psychological violence; and physical, emotional, medical or educational neglect (Supplementary Table 1). A summary list of VAC definitions used in the underlying studies is presented in Supplementary Table 2.

Based on the above criteria, we included 111 unique studies (Supplementary Table 3)—the majority being prospective cohorts (n = 69) and from high-income nations (n = 89)—from which we leveraged relevant data for those risk–outcome pairs with at least 3 studies reporting on an outcome definition consistent with the case definitions used for GBD causes (Supplementary Tables 4 and 5). These studies formed the basis for assessing 15 eligible health outcomes potentially associated with childhood physical violence (94 studies), 11 outcomes related to psychological violence (55 studies) and 9 outcomes connected to neglect (50 studies). The most frequently investigated health consequences across the various forms of VAC included mental health conditions, such as major depressive disorder and anxiety disorders; substance use disorders, such as alcohol use disorders and drug use disorders; followed by diabetes mellitus type 2, asthma, self-harm and schizophrenia (Table 2). Key analytic parameters, including pooled relative risk (RR) estimates and surrounding uncertainty—both without between-study heterogeneity/gamma included

**Table 2 | Strength of the evidence for the relationship between exposure to childhood physical violence, psychological violence and neglect and 15 health outcomes**

| Health outcome | Mean RR | 95% UI for the mean RR without gamma | 95% UI for the mean RR with gamma | BPRF | ROS | Star rating | Publication bias | Number of studies (obs.) | Selected bias covariates |
|---|---|---|---|---|---|---|---|---|---|
| **Physical violence** | | | | | | | | | |
| Major depressive disorder | 1.54 | 1.42–1.67 | 1.09–2.16 | 1.16 | 0.07 | ☆☆ | No | 26 (32) | None |
| Ischaemic heart disease | 1.45 | 1.28–1.65 | 1.10–1.92 | 1.15 | 0.07 | ☆☆ | No | 4 (5) | None |
| Alcohol use disorders | 1.54 | 1.33–1.79 | 0.96–2.46 | 1.04 | 0.02 | ☆☆ | No | 8 (17) | Effect size for male and female sexes combined |
| Eating disorders | 2.13 | 1.46–3.13 | 0.91–5.02 | 1.04 | 0.02 | ☆☆ | No | 4 (5) | None |
| Drug use disorders | 1.40 | 1.24–1.58 | 0.96–2.03 | 1.02 | 0.01 | ☆☆ | No | 11 (20) | Unadjusted for age; unadjusted for age, sex and at least one other confounding variable |
| Diabetes mellitus type 2 | 1.11 | 1.05–1.17 | 0.98–1.26 | 1.00 | −0.001 | ☆ | No | 12 (21) | None |
| Migraine | 1.41 | 1.22–1.63 | 0.93–2.13 | 1.00 | −0.002 | ☆ | No | 4 (6) | None |
| Anxiety disorders | 1.26 | 1.14–1.39 | 0.93–1.70 | 0.98 | −0.01 | ☆ | No | 11 (15) | Family/household perpetrator; exposure defined as being above the age of 15; outcome is defined as post-traumatic stress disorder |
| Self-harm | 2.00 | 1.62–2.46 | 0.86–4.66 | 0.98 | −0.01 | ☆ | No | 16 (18) | Risk of reverse causation; family/household perpetrator |
| Asthma | 1.49 | 1.22–1.82 | 0.76–2.93 | 0.85 | −0.08 | ☆ | No | 6 (8) | Unadjusted for age; unadjusted for sex |
| Gynaecological diseases | 1.20 | 1.02–1.41 | 0.69–2.06 | 0.76 | −0.14 | ☆ | No | 6 (6) | Risk of selection bias; family/household perpetrator |
| Maternal abortion and miscarriage | 1.99 | 0.99–4.00 | 0.27–14.80 | 0.37 | N/A | | No | 3 (6) | None |
| Schizophrenia | 1.96 | 0.98–3.93 | 0.24–15.88 | 0.34 | N/A | | No | 4 (6) | None |
| Sexually transmitted infections excluding HIV | 1.08 | 0.54–2.16 | 0.21–5.49 | 0.27 | N/A | | No | 3 (5) | None |
| Stroke | 1.21 | 0.85–1.74 | 0.57–2.61 | 0.64 | N/A | | No | 3 (3) | None |
| **Psychological violence** | | | | | | | | | |
| Drug use disorders | 1.38 | 1.22–1.56 | 1.03–1.85 | 1.08 | 0.04 | ☆☆ | No | 7 (10) | Risk of selection bias |
| Migraine | 1.91 | 1.49–2.43 | 0.96–3.79 | 1.07 | 0.03 | ☆☆ | No | 3 (5) | None |
| Gynaecological diseases | 1.28 | 1.15–1.43 | 0.98–1.67 | 1.02 | 0.01 | ☆☆ | No | 3 (3) | None |
| Diabetes mellitus type 2 | 1.13 | 1.05–1.23 | 0.91–1.41 | 0.95 | −0.03 | ☆ | No | 7 (13) | Unadjusted for sex |
| Schizophrenia | 2.68 | 1.71–4.22 | 0.75–9.64 | 0.92 | −0.04 | ☆ | No | 5 (6) | None |
| Major depressive disorder | 1.93 | 1.51–2.48 | 0.61–6.13 | 0.73 | −0.15 | ☆ | No | 17 (22) | Unadjusted for sex: exposure defined including ages above 15; effect size for male and female sexes combined; risk of reverse causation; unadjusted for age; family/household perpetrator: males included in the effect size |
| Asthma | 2.01 | 1.36–2.99 | 0.59–6.83 | 0.72 | −0.16 | ☆ | No | 4 (5) | Unadjusted for age |
| Anxiety disorders | 2.01 | 1.33–3.04 | 0.45–8.90 | 0.58 | −0.27 | ☆ | No | 7 (9) | None |
| Alcohol use disorders | 2.14 | 1.37–3.34 | 0.42–10.92 | 0.54 | −0.31 | ☆ | No | 7 (10) | Family/household perpetrator; unadjusted for age, sex and at least one other confounding variable; males included in the effect size |
| Self-harm | 3.08 | 1.68–5.65 | 0.27–34.48 | 0.40 | −0.45 | ☆ | No | 8 (8) | Representativeness; risk of reverse causation; risk of selection bias; unadjusted for age; exposure defined including ages above 15 |
| Ischaemic heart disease | 1.69 | 0.98–2.93 | 0.34–8.32 | 0.44 | N/A | | No | 3 (4) | None |
| **Neglect** | | | | | | | | | |
| Anxiety disorders | 1.43 | 1.28–1.59 | 1.10–1.85 | 1.15 | 0.07 | ☆☆ | No | 9 (14) | None |
| Asthma | 1.47 | 1.17–1.86 | 0.80–2.72 | 0.88 | −0.06 | ☆ | No | 3 (7) | None |

**Table 2 (continued) | Strength of the evidence for the relationship between exposure to childhood physical violence, psychological violence and neglect and 15 health outcomes**

| Health outcome | Mean RR | 95% UI for the mean RR without gamma | 95% UI for the mean RR with gamma | BPRF | ROS | Star rating | Publication bias | Number of studies (obs.) | Selected bias covariates |
|---|---|---|---|---|---|---|---|---|---|
| Major depressive disorder | 1.60 | 1.33–1.92 | 0.74–3.45 | 0.84 | −0.09 | ☆ | No | 17 (27) | Risk of selection bias; unadjusted for sex; unadjusted for age, sex and at least one other confounding variable; males included in the effect size |
| Schizophrenia | 2.49 | 1.08–5.72 | 0.22–28.78 | 0.32 | −0.57 | ☆ | No | 4 (5) | None |
| Self-harm | 2.41 | 1.22–4.75 | 0.20–29.52 | 0.29 | −0.61 | ☆ | No | 7 (8) | Risk of reverse causation; family/household perpetrator |
| Drug use disorders | 2.28 | 1.11–4.68 | 0.19–27.21 | 0.28 | −0.63 | ☆ | No | 5 (8) | Exposure defined including ages above 15; exposure defined as below age 15; representativeness; risk of selection bias |
| Alcohol use disorders | 1.84 | 0.82–4.15 | 0.11–29.67 | 0.18 | N/A | | No | 5 (9) | Unadjusted for age, sex and at least one other confounding variable |
| Diabetes mellitus type 2 | 1.07 | 0.93–1.23 | 0.72–1.60 | 0.76 | N/A | | No | 6 (14) | None |
| Sexually transmitted infections excluding HIV | 1.13 | 0.93–1.39 | 0.71–1.81 | 0.76 | N/A | | No | 3 (9) | None |

The reported mean RR reflects the risk an individual who has experienced childhood physical violence, psychological violence or neglect has of developing the health outcome in the corresponding row relative to that of someone who has not been exposed to the given form of violence. The 95% UI for the mean RR without gamma refers to the 95% UI that is estimated without fully incorporating between-study heterogeneity, while the 95% UI for the mean RR with gamma refers to the 95% UI that is estimated fully incorporating between-study heterogeneity and the uncertainty around quantified between-study heterogeneity. As the conservative estimate of excess risk consistent with existing evidence, it corresponds to the fifth quantile RR estimate incorporating between-study heterogeneity closest to the null value of 1. The percentage of excess risk is derived as (BPRF−1)×100. The ROS is derived as the signed natural log(BPRF)/2. It is calculated for risk–outcome pairs that were found to be statistically significant when estimating a conventional RR and 95% UI (95% UI without gamma) and reflects the strength of the association and available evidence. It is translated into a star rating ranging from 1 (weak evidence) to 5 (strong and consistent evidence of a strong association) based on predetermined thresholds documented elsewhere. Risk–outcome pairs that do not meet the standard for calculating the ROS are not assigned a star rating and are marked with an N/A (not applicable) ROS. The risk of publication bias is flagged on the basis of the results of Egger's regression and should inform the interpretation of the results. The selected bias covariates were flagged as reflecting statistically significant sources of systematic bias and were adjusted for in the final models. obs., number of observations informing each model pulled from the included studies.

in uncertainty estimates, aligning with traditional meta-analyses, and with between-study heterogeneity included, more fully capturing the heterogeneity across underlying studies—are reported for each risk–outcome relationship in Table 2. We also report a Burden of Proof Risk Function (BPRF) value and corresponding Risk–Outcome Score (ROS) and star rating for each pair. The BPRF (equivalent to the fifth percentile of RR draws closest to null) complements the mean RR function and, by design, reflects the lowest reasonable estimate of the effect of the violence exposure on the health outcome of interest, based on the currently available evidence. The conservative BPRF is converted into a ROS, with higher, more positive ROS values indicating a greater harmful effect supported by strong underlying evidence. These ROS values are translated into corresponding increasing star ratings, ranging from 1 to 5, to simplify the interpretation of results[29]. Pooled RR and star ratings for each risk–outcome are also displayed in Fig. 2 for ease of comparison across forms of violence and health outcomes.

**Physical violence exposure**

Among the 15 analysed health outcomes, mental health and substance use disorders made up the bulk of the 5 outcomes with the highest-rated (2 stars; ROS between the 0.00 and 0.14 predefined thresholds[29]) associations with exposure to physical VAC. Twenty-six studies[31–56] examined the association between childhood physical violence and major depressive disorder (Fig. 3 and Supplementary Table 3). We conservatively estimated that those who experienced physical violence as a child have at least a 16% higher risk of major depressive disorder compared with their unexposed counterparts (BPRF 1.16). The star rating of two indicates moderately weak evidence of an association, due to either or both a small effect size and lack of consistent evidence (Table 2). Similarly, the association of physical violence during childhood with alcohol use disorders[48,57–63], eating disorders[45,53,64,65] and drug use disorders[57,59,60,62,63,66–71] were also rated as moderately weak (two

stars), with a minimum increase in risk of 4% (BPRF 1.04), 4% (BPRF 1.04) and 2% (BPRF 1.02), respectively, for each of these outcomes in people who experienced physical violence as a child (Table 2). Our findings across these outcomes are largely, but not entirely, consistent regardless of model parameters and data point restrictions tested in various sensitivity analyses (Extended Data Fig. 1, Supplementary Tables 6–9 and Supplementary Figs. 1–7). Among the most notable differences, we identified a three-star association between childhood physical violence and alcohol use disorders when the outcome definition was restricted to clinical alcohol dependence instead of also including measures of alcohol abuse (Extended Data Fig. 1 and Supplementary Table 8).

Other than mental health and substance use disorders, ischaemic heart disease (IHD)[58,72–74] accounted for the remaining two-star association with childhood physical violence. We estimated a 15% minimum increase in the risk of this condition for individuals exposed to physical violence in childhood (BPRF = 1.15). The association was reduced to a one-star rating when including only effect sizes that examined physical violence committed by an unspecified or unrestricted perpetrator (Extended Data Fig. 1 and Supplementary Table 10).

Anxiety disorders[31,35,40,43,47,51,56,75–78] (Fig. 3 and Supplementary Table 11) was the only mental health condition among the six outcomes with one-star associations with childhood physical violence (Table 2). While we identified sufficient evidence of an association for this risk–outcome pair, the evidence was rated as weak. The star rating for this association changed to two stars when excluding studies that only consisted of post-traumatic stress disorder in their outcome definitions and when excluding perpetrator-specific violence exposures (Extended Data Fig. 1 and Supplementary Table 11).

Similarly, we found weak one-star evidence of associations with childhood physical violence for type 2 diabetes[41,58,74,79–87], migraine[58,88–90], self-harm[37,40,48,53,91–102], asthma[58,81,103–106] and gynaecological diseases[107–112]. The high degree of variability between included

studies, low reported effect sizes and low numbers of included studies for each of these risk–outcome pairs contributed to the lower strength of evidence and wider uncertainty intervals (UIs) incorporating gamma (Table 2). Our findings across these risk–outcome pairs are largely robust to various sensitivity analyses (Extended Data Fig. 1, Supplementary Tables 12–15 and Supplementary Figs. 1–7). However, for gynaecological diseases, our findings change when we focus solely on observations with non-specific perpetrators. In this case, the star-rating framework is no longer applicable because the conventional estimate of 95% uncertainty (without gamma) around the mean RR curve encompasses the null value of an RR of 1, indicating insufficient evidence of an association with childhood physical violence, potentially due to the low number of included observations (Extended Data Fig. 1 and Supplementary Table 15). For type 2 diabetes and self-harm, the strength of evidence rating shifted from one to two stars when we did not apply 10% trimming to the data and when running perpetrator-stratified models, respectively (Extended Data Fig. 1 and Supplementary Tables 12 and 13).

Lastly, maternal abortion and miscarriage[113–115], schizophrenia[116–119], sexually transmitted infections (STIs) excluding human immunodeficiency virus/acquired immunodeficiency syndrome (HIV/AIDS)[120–122], and stroke[81,123,124] were outcomes where we found insufficient evidence of an association with physical violence (Table 2). For STIs, the three included studies reported on a range of infections, including syphilis, chlamydia and genital herpes (Supplementary Table 3). The findings for STIs and abortion/miscarriage are robust to removing the standard error adjustment for overlapping analytical samples (Supplementary Tables 16 and 17), while this change resulted in finding a one-star statistically significant association for schizophrenia (Extended Data Fig. 1 and Supplementary Table 18).

## Psychological violence exposure

Among the 11 health outcomes analysed in relation to childhood exposure to psychological violence (Fig. 4), drug use disorders, migraine and gynaecological diseases were two-star risk–outcome pairs, indicating moderately weak evidence of an association (Table 2). This category of violence encompasses being verbally threatened, sworn at, insulted or witnessing other forms of domestic violence (Supplementary Table 2). Our findings indicate that experiencing psychological violence in childhood leads to at least an 8% increase in the risk of developing drug use disorders[57,62,67,68,70,125,126] (BPRF = 1.08; Table 2). Notably, we identified a three-star association (moderate evidence of association) between childhood psychological violence and drug use disorders when the outcome definition was restricted to outcomes related to illicit drugs other than marijuana (Extended Data Fig. 2, Supplementary Table 7 and Supplementary Fig. 8).

The association between exposure to psychological violence and migraine[58,89,90] was found to have a BPRF of 1.07, suggesting a conservative estimate of at least a 7% increased risk of this condition for individuals exposed to psychological violence in childhood. For gynaecological diseases[109,111,112], we estimated at least a 2% higher risk of these conditions for women exposed to psychological violence as a child (BPRF = 1.02; Table 2).

Beyond these three risk–outcome pairs, type 2 diabetes mellitus[58,74,81,82,85–87], schizophrenia[116,117,119,127,128], major depressive disorder[31,34,36,38,42–44,48–53,55,56,129,130], asthma[58,81,104,106], anxiety disorders[31,43,51,56,76,78,131], alcohol use disorders[48,57,58,61,62,70,132] and self-harm[48,53,93,94,97,100–102] were all found to have one-star associations with exposure to psychological violence during childhood. This star rating indicates that, while the existing data point to a statistically significant relationship, the strength of this evidence is weak and subject to change with additional evidence. These findings were largely robust to the feasible sensitivity analyses (Extended Data Fig. 2, Supplementary Tables 6, 8, 11–14 and 18 and Supplementary Figs. 8–14). However, the star rating for the relationship between childhood psychological violence and major depressive disorder increased from

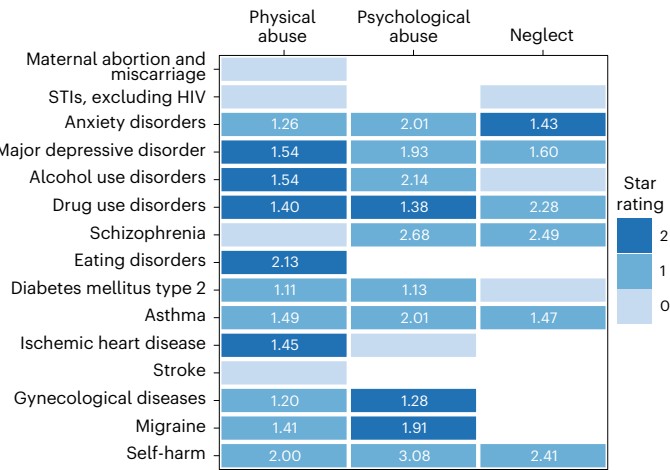

**Fig. 2 | Mean RR and strength of the evidence for the association between multiple forms of VAC and 15 health outcomes.** The number depicted in each box corresponds to the mean RR estimated as part of this study for associations between physical violence, psychological violence, neglect during childhood and specific health outcomes. Coloured boxes represent all associations supported by at least three published studies, allowing our Burden of Proof assessment, while empty white cells reflect risk–outcome pairs for which we did not have enough data (three or more studies) to examine. The shades of the blue boxes represent the strength of evidence supporting each association based on our conservative interpretation of the data that aligns with the Burden of Proof approach. The strength of the evidence is translated into a star rating from 1 to 5 stars based on thresholds outlined in Zheng et al.[29], where one star denotes weak evidence and each additional star indicates progressively stronger evidence. An absence of stars (zero stars) signifies insufficient evidence of a statistically significant association between the exposure and the outcome.

one star to three stars when the analysis was limited to family/household perpetrator-specific observations and to two stars when using only women-specific observations or omitting studies that refer to depression, broadly, as their outcome (Extended Data Fig. 2, Supplementary Table 6 and Supplementary Figs. 9–11). Similarly, the star rating increased to two stars for schizophrenia and type 2 diabetes when perpetrator-specific observations were excluded and when the analysis was restricted to women, respectively (Extended Data Fig. 2, Supplementary Tables 12 and 18 and Supplementary Figs. 9 and 12). When omitting perpetrator-specific observations, there was insufficient evidence of an association between this form of violence and alcohol use disorder and type 2 diabetes, and for the latter, this was further true when removing automatic trimming and restricting the outcome definition (Extended Data Fig. 2, Supplementary Tables 8 and 12 and Supplementary Figs. 9, 13 and 15).

Finally, we found insufficient evidence of an association between psychological violence during childhood and IHD[58,73,74] based on the existing literature. The three included studies with data on IHD (Supplementary Table 3) yielded four observations with a high degree of between-study heterogeneity (gamma = 0.20 (s.d. = 0.19); Supplementary Table 19).

## Exposure to neglect

We identified nine health outcomes with three or more studies reporting on their association with some form of childhood neglect, including emotional and/or physical neglect by caregivers or other adult figures (Fig. 5 and Supplementary Table 3). Mental health and substance use disorders again made up the majority of these health outcomes with 17 studies included for major depressive disorder[32,35–38,41–44,49–52,129,133–135], 9 for anxiety disorders[35,43,51,75–78,134,135], 5 for drug use disorders[67,71,125,126,136] and for alcohol use disorders[57,58,61,132,137], and 4 for schizophrenia[117,119,128,138]. Drawing upon the eligible observations,

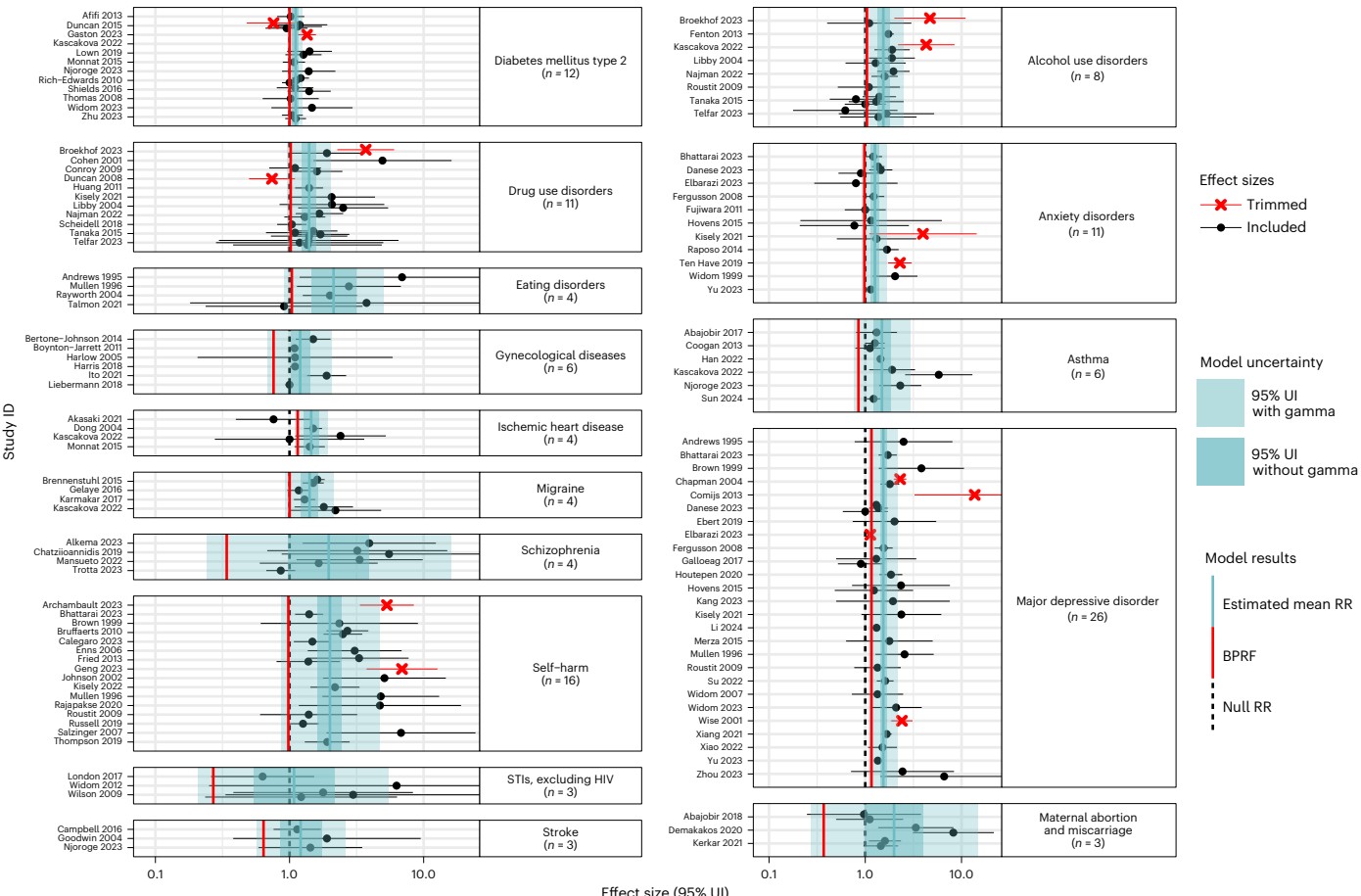

**Fig. 3 | Forest plots for physical violence during childhood and health outcomes identified through a systematic review of the literature.** These forest plots present the estimated pooled RR, its 95% UIs and the data points underlying the estimates for 13 health outcomes studied in association with childhood physical violence. Each data point and horizontal line corresponds to a mean effect size and 95% UI from the included study identified on the *y* axis. The colour of the point indicates whether the point was detected and trimmed as an outlier. The light-blue interval corresponds to the 95% UI of the pooled RR when incorporating between-study heterogeneity; the dark-blue interval corresponds to the 95% UI of the pooled RR without between-study heterogeneity. The vertical dotted black line reflects the null RR value (RR = 1), and the vertical red line is the Burden of Proof function at the fifth quantile for this harmful risk–outcome association. We truncated the *x* axis to make the scale more legible, so a handful of 95% UIs from the included studies extend beyond the plot margin. The full uncertainties are presented in Supplementary Table 34. We included multiple observations from a single study when effects were reported by severity/frequency of the violence exposure, by different types of violent acts and/or separately by sex or other subgroups.

we found a statistically significant association between childhood neglect and all but one (alcohol use disorders) of the outcomes related to mental health and substance use disorders (Table 2). Anxiety disorders were found to have the highest BPRF and ROS at 1.15 and 0.07, respectively, which translates into a two-star rating (moderately weak evidence of association) and an estimated conservative minimum of 15% increased risk of anxiety disorders among individuals who experienced neglect during their childhood (Table 2). Major depressive disorder, schizophrenia and drug use disorders were found to have one-star associations with childhood neglect (Table 2). While these findings were robust across a wide range of sensitivity analyses (Extended Data Fig. 3, Supplementary Tables 6, 7 and 18 and Supplementary Figs. 16–22), restricting data to women-only samples resulted in a lack of evidence for an association between childhood neglect and both major depressive disorder and drug use disorders (Extended Data Fig. 3, Supplementary Tables 7 and 8 and Supplementary Fig. 16). Men-only samples also resulted in insufficient evidence of an association between childhood neglect and drug use disorders, while being more restrictive in the outcome definitions accepted for major depressive disorder resulted in a two-star association (Extended Data Fig. 3, Supplementary Tables 7 and 8 and Supplementary Figs. 17 and 20).

Among the other four health outcomes that were feasible to analyse, asthma[58,104,106] and self-harm[37,93–95,97,101,102] were both found to have weak one-star associations with being neglected during childhood. Results for asthma were robust to removing the standard error adjustment, while self-harm persisted as a one-star association when excluding perpetrator-specific observations and dropped to insufficient evidence of an association when looking only at family/household perpetrators (Extended Data Fig. 3, Supplementary Tables 13 and 14 and Supplementary Figs. 17 and 20). Consistent with the findings related to childhood physical abuse, we found insufficient evidence to suggest an association between exposure to neglect and subsequent STIs[121,122,139]. We also found insufficient evidence of an association between childhood neglect and type 2 diabetes[41,58,82,85,87,140] (Table 2). These findings for STIs and diabetes remained consistent in feasible sensitivity analyses (Extended Data Fig. 3, Supplementary Tables 12 and 17 and Supplementary Figs. 15–21).

## Discussion

The present study evaluates the adverse health consequences associated with exposure to physical violence, psychological violence and neglect during childhood. Using the Burden of Proof analytic

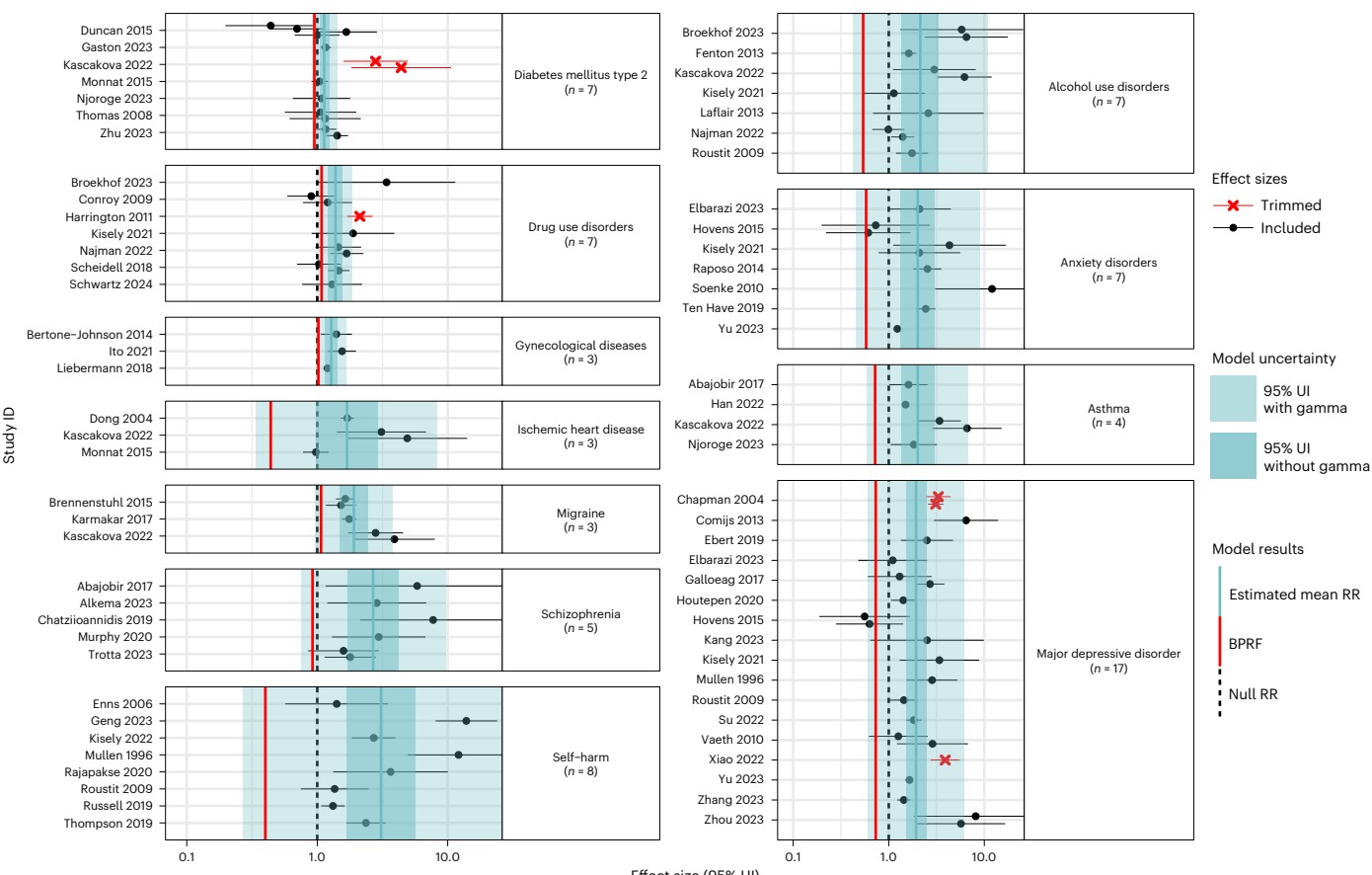

**Fig. 4 | Forest plots for psychological violence during childhood and health outcomes identified through a systematic review of the literature.** These forest plots present the estimated mean RR, its 95% UIs and the data points underlying the estimates for ten health outcomes studied in association with childhood psychological violence. Each data point and horizontal line corresponds to a mean effect size and 95% UI from the included study identified on the *y* axis. The colour of the point indicates whether the point was detected and trimmed as an outlier. The light-blue interval corresponds to the 95% UI of the pooled RR when incorporating between-study heterogeneity; the dark-blue

interval corresponds to the 95% UI of the pooled RR without between-study heterogeneity. The vertical dotted black line reflects the null RR value (RR = 1), and the red vertical line is the Burden of Proof function at the fifth quantile for this harmful risk–outcome association. We truncated the *x* axis to make the scale more legible, so a handful of 95% UIs from the included studies extend beyond the plot margin. The full uncertainties are presented in Supplementary Table 34. We included multiple observations from a single study when effects were reported by severity or frequency of the violence exposure, by different types of violent acts and/or separately by sex or other subgroups.

framework, we generated intentionally conservative assessments of the associations between exposure to these forms of violence and negative health outcomes based on methods that accounted for both known variability across study-design characteristics and remaining heterogeneity among input-level effect estimates, yielding highly robust and credible findings. In relation to childhood physical violence, our analysis found statistically significant increased risks for 11 of the 15 health outcomes analysed for individuals exposed to this form of violence. Associations between physical violence and major depressive disorder, IHD, substance use disorders and eating disorders received two-star (moderately weak) ratings, a measure that summarizes both the strength of the association and the strength of the underlying evidence. Similarly, we found statistically significant associations between childhood psychological violence and 10 of 11 health outcomes analysed, with the associations to drug use disorders, migraine and gynaecological diseases also receiving two-star ratings. We also identified statistically significant associations between childhood neglect and six of nine health outcomes analysed, with the relationship with anxiety being rated two stars. Particularly given our conservative interpretation of the evidence, the adverse health consequences found by our analysis in association with VAC—which had already been underscored by some traditional meta-analytical efforts[141,142]—reinforces the need for public

health approaches to encourage the prevention of all forms of childhood violence, to better address the health needs of survivors and to bolster the evidence base relating health consequences to childhood violence exposure.

According to our results, one area of considerable increased risk associated with VAC is mental disorders. Consistent with the literature, we found that all forms of violence were associated with greater risks for major depressive disorder and anxiety disorders[9,143,144]. Under the conservative interpretation of the available evidence of the Burden of Proof approach, CSA, presented in a previous publication[10], and physical violence were found to increase the risk of major depressive disorder by at least 20% and 16%, respectively, while exposure to neglect during childhood was found to increase the risk of anxiety by at least 15%. This is particularly alarming given that mental disorders are the second leading cause of disease burden worldwide among individuals aged 5–14 and the leading cause for 15–49-year-olds[145]. Notably, in our analysis, we adhered to the GBD definitions of depression and anxiety, ensuring that only studies with rigorous mental health diagnosis definitions were included. This approach represents an improvement over previous scientific efforts that captured self-reported depressive or anxious symptoms, without diagnosis, as part of their outcome definition[25,146].

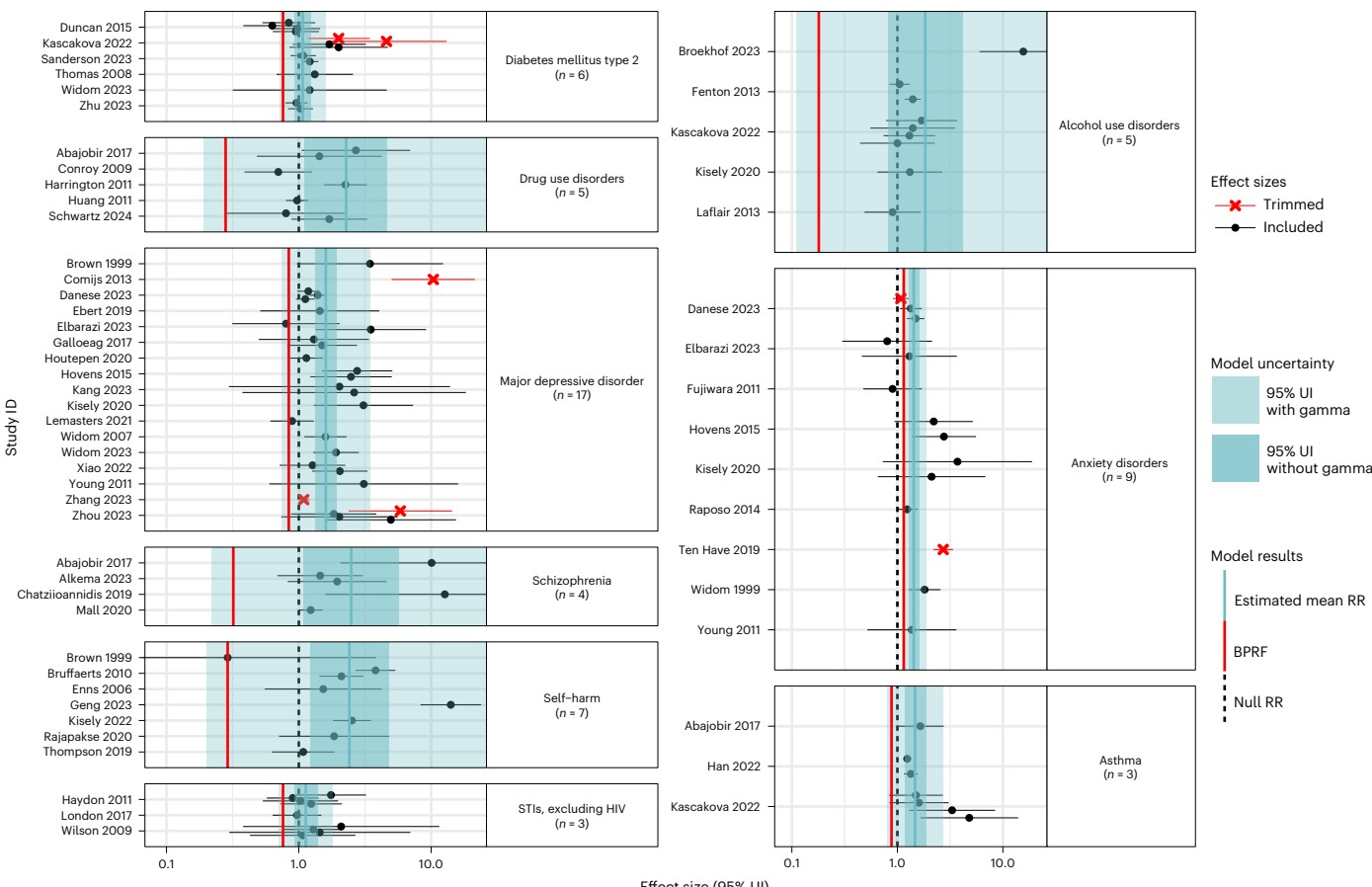

**Fig. 5 | Forest plots for neglect during childhood and health outcomes identified through a systematic review of the literature.** These forest plots present the estimated mean RR, its 95% UIs and the data points underlying the estimates for nine health outcomes studied in association with childhood neglect. Each data point and horizontal line corresponds to a mean effect size and 95% UI from the included study identified on the *y* axis. The colour of the point indicates whether the point was detected and trimmed as an outlier. The light-blue interval corresponds to the 95% UI of the pooled RR when incorporating between-study heterogeneity; the dark-blue interval corresponds to the 95% UI of the pooled RR without between-study heterogeneity. The vertical dotted black line reflects the null RR value (RR = 1), and the vertical red line is the Burden of Proof function at the fifth quantile for this harmful risk–outcome association. We truncated the *x* axis to make the scale more legible, so a handful of 95% UIs from the included studies extend beyond the plot margin. The full uncertainties are presented in Supplementary Table 34. The 95% UI that incorporates between-study heterogeneity for our estimate of the mean RR of schizophrenia is also truncated here and presented in Table 2. We included multiple observations from a single study when effects were reported by severity or frequency of the violence exposure, by different types of violent acts and/or separately by sex or other subgroups.

Recent meta-analytical efforts have also showcased substance use disorders as another important category of increased health risks, with age-, gender- and violence-type-dependent associations to childhood violence exposure[17,147,148]. When accounting for known and unknown sources of between-study variation and potential bias, we found the risk of alcohol use disorders escalating by a minimum of 45% and 4% following childhood experiences of sexual violence[10] and physical violence, respectively. While here we assess the impact of VAC on alcohol use that meets the criteria for an alcohol use disorder—a medical condition characterized by an impaired ability to stop or control alcohol use—extensive literature also points to an association between different forms of violence and varying levels of alcohol consumption[149,150]. Given that even small amounts of alcohol increase the risk for a wide array of diseases[151] and physical abuse and domestic VAC are greater in households where someone drinks alcohol excessively, there is a pressing need for alcohol policies and interventions that focus specifically on preventing health harms and stopping the perpetuation of the cycle of violence[152].

Although our Burden of Proof findings provide confirmation of the adverse health consequences associated with each form of VAC and despite the pooled effect sizes as large as 3.08, most of the statistically significant risk–outcome relationships identified in our study received moderately weak or weak star ratings, mirroring the findings for CSA[10]. This is primarily due to the weak and/or inconsistent evidence base, that is, small numbers of input studies and high degree of between-study heterogeneity. Out of the 35 risk–outcome pairs we assessed, almost half of our analyses were based on five or fewer studies, and a considerable portion met only the minimal threshold of three studies required for evaluation. The paucity of research, particularly on the psychological violence and neglect dimensions of VAC, poses challenges for comparing the negative impacts of different forms of violence across the same health outcomes. Anxiety disorders, major depressive disorder, alcohol use disorders, drug use disorders, schizophrenia, type 2 diabetes mellitus, asthma and self-harm were the only outcomes with sufficient data to analyse associations with all categories of childhood violence, including sexual[10], physical and psychological violence, and neglect. This gap in available evidence highlights an urgent need to invest in well-designed studies from a diverse set of countries and contexts that can elucidate the full spectrum of VAC's health implications and how survivors may face the consequences of violence long after their experiences. As highlighted by the World Health Organization action plan to strengthen health systems' capacity

to address abuse against children[153], timely and compelling evidence detailing the burden of VAC is critical for tailoring effective prevention and response strategies.

While our research concentrates on the health impacts of VAC, the literature robustly demonstrates the correlation between all forms of childhood violence and other adverse lifelong consequences, such as educational deficits[18,154], increased likelihood of juvenile delinquency and criminal behaviour[8,155], decreased economic success in adulthood[19] and heightened risk of girls and women becoming victims of violence and boys and men perpetrating violence later in life[156,157]. These wide-ranging indirect repercussions of VAC point to the critical need for persistent, collaborative and intersectoral approaches in the detection, monitoring, prevention and intervention processes aimed at shielding children from harm and guaranteeing their right to safety and to a future that is safe, long, healthy, fulfilling and productive.

Importantly, breaking the cycle of violence requires a comprehensive approach that goes beyond just addressing abuse of children to additionally be responsive to gender dynamics and to tackle violence across the entire lifespan. The interconnection between VAC and violence in adulthood is indicated by studies revealing that witnessing violence as a child can increase the likelihood of involvement in child maltreatment and subsequent domestic violence—both as perpetrator and as victim[20,158]. Consequently, it is vital to address the shared risk factors associated with both child and adult abuse, as these types of violence frequently coexist within the same environments and are influenced by overlapping social and economic factors[159,160].

The interpretation of the findings of our study should take into account several limitations. First, there is important heterogeneity in the definitions and measurement of VAC across the studies included. VAC is a multifaceted issue that can be characterized by the nature of the violent act, the impact on the victim, the relationship between the victim and the perpetrator(s), and the context in which the violence occurs. There are also substantial challenges when measuring these characteristics, including potential recall bias or dissociative amnesia when self-reporting exposure and the underreporting of violence through formal channels, which can affect the magnitude of the associations identified[161,162]. To address the challenge of synthesizing heterogeneous data and enhance the comparability of our findings with future research, we operationalized the nature of violence in accordance with the ICVAC[30]. Furthermore, to mitigate the influence of differing definitions related to specific categories of perpetrators, we relied on bias covariates. However, our ability to control for other dimensions of violence exposure and assessment was constrained by the dearth of data and, in some cases, by the absence of a clear gold-standard practice for exposure assessment. For example, both prospective (in the form of formal reports) and retrospective assessments of childhood violence have substantial weaknesses that may result in underreporting or skewed reporting. Consequently, this limitation may have contributed to the unexplained between-study heterogeneity estimated as part of the Burden of Proof methodology.

Second, our analysis does not capture the entire scope of violence that children, particularly girls, are subjected to. The scope of our study is limited by the availability of data and, as a result, does not capture critical forms of violence such as female genital mutilation (FGM), child marriage, bullying and technology-facilitated violence[163–165]. FGM is a deeply entrenched practice, particularly in various African cultures, offering no health benefits and leading to a host of complications for girls, women and their families[166]. The dearth of high-quality research on the health effects of FGM is an important barrier to understanding its full impact. Similarly, technology-facilitated violence is a relatively new and growing concern, posing unique challenges that are not yet fully understood and underscore the need for further research[167].

Furthermore, our selection of risk–outcome pairs was constrained to outcomes reported by three or more relevant studies in association with distinct forms of violence. Consequently, we were unable to evaluate the association between VAC and a number of health outcomes for which some evidence of association exists, such as chronic pain[168], multiple sclerosis[169] and other chronic respiratory conditions[170,171]. In addition, our analysis focused on reports of distinct forms of child abuse, and it did not account for the co-occurrence of violence types that often take place together[172,173]. Co-occurrence of different forms of violence may act as compounding risk factors[174], particularly when considering the outcomes we found to be shared across physical and psychological violence and neglect, such as major depressive disorder. Unfortunately, we found that most studies did not explicitly state whether their case definitions were exclusive to the type of violence in question, if they also included children who had concurrently experienced other forms of violence or if they had even investigated the possibility of polyvictimization. On occasion, a study provided multiple data points accounting for different forms of violence. In these cases, we selected the data points that were explicitly restricted to individuals who had only experienced the violence type in question or that controlled for exposure to another form of violence in order to narrow in on the distinct health toll of different types of VAC. While there were insufficient data on specific combinations of violence to fully explore the consequences of co-occurrence, survivors often experience multiple forms of VAC, and these health risks may compound over multiple exposures. As a result, our approach presents only the lowest bound of associated health risk a survivor may experience in the context of polyvictimization. Future research must extend the scope of both violence exposures and health outcomes examined. Only by fully depicting the breadth of VAC's extensive health implications can global action ultimately be driven to protect children and mitigate the effects of violence.

Third, another limitation of our study is that we were unable to disaggregate the effects of violence by age of exposure, ethnicity or other demographic variables owing to the constraints of the available data. This is a critical area for future research, as some groups of children may be at an increased risk of violence. Understanding differential impacts, both among high-risk groups and across different epigenetic contexts[175], is essential for tailored interventions. Moreover, by not being able to account for different epigenetic contexts, we must consider the possible confounding role of genetics, which may increase familial risk of mental or substance use disorders, such as alcohol use disorder, and, in turn, may increase the risk of violence exposure[176–178]. In lieu of more detailed data that account for genetic factors, these associations may exacerbate some of our observed effects between VAC and conditions with genetic components. Moreover, the preponderance of our data originated in high-income countries. This geographic bias suggests the need for more diverse studies to assess the health impacts of VAC comprehensively. Expanding the geographic representation of research to incorporate low- and middle-income countries could inform culturally specific protective measures for more effective policy responses. Lastly, our study conceptualizes physical violence, psychological violence and neglect as dichotomous risks. This approach, which arises from the limited detail in the available data, may not capture the complexities of violence exposure, overlooking the intricacies of timing, frequency and severity of violent experiences. Recognizing this limitation, it is essential for future research to utilize longitudinal and linked datasets to more accurately assess the cumulative health effects of various types of violence. Such research would provide insights into the nuanced impacts of violence experienced at different developmental stages and frequencies, enhancing our understanding of the lifelong compounding effects of violence.

In conclusion, this systematic review illuminates the extensive negative health impacts of childhood physical violence, psychological violence and neglect. Our findings demonstrate that VAC places children at an elevated risk of enduring a wide array of health consequences, including but not limited to mental health and substance use disorders, injuries, chronic conditions and reproductive health problems. The

evidence generated by this study supports the reevaluation and broadening of the scope of childhood violence forms considered within GBD, as our results confirm that every form of VAC has a substantial health impact. A thorough representation of the health implications of VAC is critical for informing policy, mobilizing resources and propelling the global agenda towards eliminating VAC and reducing its extensive societal and individual burdens. Therefore, our study constitutes a pivotal call to action for governments, non-governmental organizations and stakeholders to unite in fostering adversity-free childhoods and to prioritize a renewed commitment towards achieving Sustainable Development Goal Target 16.2. Ending all forms of VAC represents both a moral imperative and a strategic investment essential for the health and prosperity of future generations.

## Methods

### Overview

In this research, we adopted the Burden of Proof framework, an approach pioneered by Zheng and colleagues[29], to conservatively estimate the associations between exposure to different forms of VAC—namely physical violence, psychological violence and neglect—and numerous health outcomes while also evaluating the strength of evidence underlying each of these associations. This methodology utilizes a meta-regression tool—Bayesian, regularized, trimmed (MR-BRT)—to derive pooled RR estimates and associated uncertainty. These methods improve estimation accuracy by detecting and trimming potentially distorting outliers in the input data using a robust, likelihood-based approach and by rigorously testing and adjusting for bias covariates that capture systematic variability across known input study characteristics. In addition, they quantify remaining heterogeneity across input studies that is otherwise not explained, which is incorporated into uncertainty estimates serving as the basis for generating a BPRF, ROS and star rating, all reflective of both the strength of the association and the strength of the underlying evidence. These measures of evidence strength generated by the Burden of Proof approach provide a crucial complement to existing approaches that evaluate strength of evidence on the basis of expert judgement[179]. These methods have already been used to evaluate health impacts linked with CSA and intimate partner violence[10], active[180] and passive smoking[181], chewing tobacco[182], high systolic blood pressure[183] and intake of unprocessed red meat[184] and vegetables[185]. Our analysis further expands the Burden of Proof literature to include additional forms of VAC, producing estimates using the same metrics and, thus, directly comparable to prior Burden of Proof findings, establishing VAC as a health risk factor akin to smoking and high systolic blood pressure. We provide below a description of the methodological steps taken to estimate the associations between adverse health outcomes and our exposures of interest, each treated as a binary risk factor (Supplementary Table 20).

For each risk–outcome assessment, we produced a set of estimates—pooled RRs and accompanying UIs, and BFRF and ROS values—drawing upon all eligible data, meaning that our primary results were not location, sex or age specific. We generated additional sex-specific results as part of our sensitivity analyses to explore potential differences in risk by sex. The present study complies with the Preferred Reporting Items for Systematic Reviews and Meta-Analyses (PRISMA) guidelines[186] (Supplementary Tables 21 and 22) and Guidelines for Accurate and Transparent Health Estimates Reporting (GATHER) recommendations[187] (Supplementary Table 23).

### Systematic review

This study is part of a wider initiative aiming to identify and synthesize all available data on the health impacts of exposure to any form of violence against women, GBV and violence against children and young people, contributing to the efforts of the Lancet Commission on Gender-Based Violence and Maltreatment of Young People. The analyses of the health effects of exposure to intimate partner violence among women and CSA, which are also components of this broader project, have been previously published[10]. The systematic review from which we drew the inputs for the present investigation was conducted in line with a prospectively published protocol (PROSPERO: CRD42022299831)[188].

In brief, we conducted a systematic search across seven databases (PubMed, Embase, CINAHL, PsycINFO, Global Index Medicus, Cochrane and Web of Science Core Collection) for all relevant studies published between 1 January 1970 and 31 January 2024. Our search strings have been previously published[10,188] and are also included in Supplementary Information section 7.1. Across the databases, we identified 75,331 unique articles reporting on any form of GBV or VAC and a variety of health outcomes for title and abstract screening. Studies that met our inclusion criteria related to study design, generalizability, GBV- or VAC-related exposure definitions and the presence of necessary data on an estimate of association[188] (Supplementary Information section 7.2) were moved to full-text screening. Studies were not eligible for inclusion if the study design did not make it feasible to establish temporality or focused on a specific and ungeneralizable population subgroup, or if the study reported only aggregate measures of exposure including non-GBV or VAC, or did not report other necessary data[188]. By reviewing the studies cited in the systematic reviews and meta-analyses captured in our searches, we identified additional references. In total, 638 cohort, case–control and case-crossover studies reporting health impacts of any form of VAC or GBV across any ages were extracted using a modified Covidence 2.0 extraction template (Supplementary Table 26). As part of the data extraction, we collected variables corresponding to study metadata, population and sample characteristics, exposure and outcome assessment, effect sizes and uncertainty estimates, and potential sources of systematic bias.

From this larger pool of studies, for the present analysis, we specifically selected the studies that examined the association between health outcomes and exposure to physical violence ($n = 94$), psychological violence ($n = 55$) or neglect ($n = 50$) during childhood. We define each of these forms of violence exposure in accordance with ICVAC definitions[30], as further described below. Our detailed inclusion and exclusion criteria are available in Supplementary Information section 7.2.

### Exposure definitions

The ICVAC[30] delineates VAC as any intentional, unwanted and non-essential act, threatened or actual, against a child or against multiple children. Such acts may result in, or have a substantial likelihood of leading to, fatal outcomes, bodily injury or a spectrum of psychological distress. The ICVAC does not distinguish between VAC and childhood maltreatment but, rather, suggests that these terms are interchangeable and synonymous. In doing so, it considers both forms of maltreatment, such as neglect, and forms of direct abuse of children as violence. Within this framework, childhood exposure to violence is categorized as any exposure occurring before the age of 18. The present analyses similarly use this demarcation line of 18 years old and adopt the terminology presented in the ICVAC. The ICVAC classifies VAC on the basis of the nature of violent acts, defining each act as either the commission or omission of intentional behaviour (including neglect). Therefore, acts may encompass direct behaviours of varying natures, including physical, verbal, non-verbal or sexual actions, or they may signify a failure to act, such as in cases of child neglect. This classification does not consider the relationship between the victim and the perpetrator, nor the setting in which the violence occurs, as criteria for categorization. In adherence to the ICVAC classification and given limitations in the data explored below, we have elected to examine the different forms of violence independently, based on the assumption that, while a single individual may often be the survivor of multiple forms of VAC, the health implications may be distinct. We defined physical violence as the exertion of physical force against a child's body, which may consist of both severe and less serious assaults. Psychological violence is characterized by verbal and non-verbal acts that inflict distress, such

as terrorizing, harassing, denigrating and humiliating a child, as well as exposing the child to domestic violence or other violent scenarios. Neglect is conceptualized as the failure to provide for a child's basic physical or psychological needs, to safeguard the child from harm or to secure necessary medical, educational or other services, particularly when those responsible for the child's welfare have the means, knowledge and the ability to access such services.

## Health outcomes and data selection

Based on our systematic review, we identified all causes of disease and injury with sufficient data (three or more studies, that is, the minimum number of data points needed to reasonably evaluate strength of evidence without undue influence by a single study and its design characteristics) to inform a potential exposure–outcome association. These health outcomes have been defined following the guidelines of the GBD (Supplementary Information section 3).

Consistent with our established case definition, our input data reflect the health risks associated with exposure before the age of 18 to a violence definition that specifically aligned with only one of the violence types of interest, regardless of the violence perpetrator. Studies that only examined combined violence types (for example, physical violence and neglect) or combined health outcomes that did not align with the GBD definitions (for example, depressive and anxiety disorders together), only assessed childhood sexual violence or only covered adulthood exposure to violence were not eligible for the present analyses. Findings related to childhood sexual violence and exposure to GBV during adulthood are documented separately[10]. While we explored the feasibility of including combined forms of violence in our analysis, there were insufficient studies with comparable combined case definitions to draw reasonable conclusions regarding the patterns of compounding risk. Studies that focused only on specific violent acts or more granular outcomes were included under the applicable umbrella exposure or outcome. For example, witnessing domestic violence was included as a form of psychological violence whereas hitting by a parent was included as physical violence in alignment with the ICVAC violence categories. Similarly, herpes- and syphilis-related outcome definitions were attributed to the broader category of STIs, and endometriosis and uterine fibroids were included under the category of gynaecological diseases following GBD cause groupings. When relevant and feasible, we also conducted sensitivity analyses limiting the outcome definitions to their various alternatives (Supplementary Information section 4).

From each study, the effect sizes that adjusted for the greatest number of relevant potential confounding variables associated with a given violence type and health outcome combination were selected to form our input datasets. When multiple adjusted effect sizes were available, we implemented a further sequential data point selection process to identify those with the closest exposure and outcome definitions to our reference definitions, the least restrictive perpetrator type and the broadest sample. First, to reduce the potential impact of co-occurrent forms of violence, we prioritized analytical samples where the exposed groups were limited to individuals who had experienced only the violence type of interest. If information regarding restrictions on the case definitions was not available or no restriction was made on the basis of other exposures to violence, we selected effect sizes that were controlled for alternative exposures to violence over the effect sizes from the same study from samples that were not explicitly restricted or adjusted for alternative exposures. For studies with multiple recall periods associated with an exposure, such as one effect size for exposure to violence in the past year and another for exposure to violence at any time during childhood, we selected the observations that corresponded to the longest recall period. Furthermore, we selected the observations for outcomes that most closely matched the GBD cause definitions if a study reported on effect sizes for different outcome subtypes or groupings.

For studies that reported on perpetrator-specific violence and on general violence perpetrated by anyone or an aggregated perpetrator grouping, we selected the observations for the broadest perpetrator type for our primary analysis. We conducted sensitivity analyses for violence perpetrated by family/household members when sufficient perpetrator-specific data were available. For studies that reported on other types of subgroup analyses in addition to a primary analysis, such as study site-specific analyses, we prioritized observations from the overall sample. Lastly, we selected observations that were derived from samples of both male and female participants over those that were sex-stratified when both were available from the same study. We conducted sex-specific sensitivity analyses selecting the sex-stratified observations when feasible to examine differences in risk and available evidence by sex. Any further studies with multiple observations underwent a manual vetting process to identify sources of more nuanced differences, including differences in the outcome and exposure ascertainment (Supplementary Information section 3). For studies with multiple eligible observations following data point selection, we downweighted observations for the same risk–outcome pair that were derived from non-mutually exclusive age, sex and location study samples. In the absence of further data on the degree of overlap in the participants informing the included effect sizes, we leveraged the square root of the number of observations as an adjustment factor multiplied by the standard error. This approach is cautious and intended to avoid the well-documented challenge of overrepresenting a single study in our meta-analyses[189].

Upon finalizing the selection of data points and compiling our input datasets, we used the MR-BRT tool for conducting multiple meta-regression analyses to estimate the pooled RR of specific health outcomes for individuals exposed to different forms of childhood violence relative to their unexposed counterparts. We used a likelihood-based least trimmed squares approach[190] to identify and exclude 10% of the data points as outliers with the potential to disproportionately impact the model. This procedure is recommended for all analyses incorporating ten or more data points.

## Testing and adjusting for biases across study designs and characteristics

We used the extracted data related to individual studies' characteristics to create a series of binary covariates to capture potential sources of systematic bias, according to the GRADE[179] approach, within our input dataset. The covariates encompassed the risk of bias arising from the representativeness of the study population and the analytical sample; the potential for selection bias; the risk of reverse causation; the upper age limit of exposure as defined by the authors; varying levels of control for confounding factors; and the type of estimated measure of association used (odds ratio versus RR). Due to the nature of VAC, we formulated a specific covariate to ascertain whether the study's definition of violence was linked to a restricted perpetrator category (that is, a family or household member) and another to denote whether exposure was ascertained from administrative databases. We also generated bias covariates to distinguish both gender-specific studies and gender-specific effect sizes from studies with multiple genders. Finally, for depressive disorders, anxiety disorders and substance abuse disorders, we developed a set of bias covariates to address variations in case definitions. A detailed description of each of the bias covariates is provided in Supplementary Information section 8.1 (Supplementary Tables 27–30), and the study-specific bias covariate values are listed in Supplementary Tables 31–33.

Using Burden of Proof methods, we systematically tested the bias covariates utilizing an automated selection algorithm that applies a step-by-step technique to identify covariates that statistically affect risk estimates. In brief, potential bias covariates are sequentially ranked using a Lasso approach[191,192] and then added one-by-one—starting with the highest ranked—as interaction terms with the crude pooled RR to

the linear meta-regression model. This process is terminated as soon as a bias covariate is added that is not significantly associated with the effect size. The full mathematical model is described in detail in the Burden of Proof methods paper by Zheng et al.[29]. Covariates were tested using this approach if there were at least two data points in the model linked to each covariate value (0 being the gold standard and 1 being the alternate). The statistically significant covariates included in the linear regression were adjusted for in the final mixed-effects model used to calculate the pooled RR estimates. These covariates reflect known variation across input study characteristics that we identified as likely sources of systematic bias and therefore controlled for them in our final model.

### Quantifying between-study heterogeneity

After adjusting for the identified sources of systematic bias, we measured and accounted for the remaining unexplained heterogeneity between studies. Following Zheng et al.[29], we added a study-level random slope to the final linear mixed-effects model to capture between-study heterogeneity (hereby referred to as gamma) and quantified the uncertainty in estimating gamma using a Fisher information matrix to account for small numbers of studies[193]. To generate what are labelled in the Burden of Proof framework as 'UIs with between-study heterogeneity', the 95% quantile of gamma was incorporated into the traditional posterior UIs around the pooled RRs estimated by the linear mixed-effects model. Effect sizes accompanied by UIs that do not include between-study heterogeneity (reported in Table 2 as the RRs without gamma) are comparable to the estimates typically reported in conventional meta-analyses.

### Evaluating publication bias

Both reporting bias (selectively reporting analyses within a study) and publication bias (selectively publishing studies based on their findings) are known to influence the results of meta-analyses. To examine their possible impact in our analyses, we used Egger's Regression, a linear regression tool designed to measure the correlation between standard error and effect size[194]. Egger's Regression evaluates the degree to which data are skewed in a direction suggesting either reporting or publication bias. This examination was supplemented by visual inspection of the risk–outcome-specific funnel plots showing the residuals for included effect sizes against the standard errors (Supplementary Information section 9). In the absence of reporting bias or publication bias, we would expect to observe skewed data outside of a standard inverted funnel around zero. If reporting or publication bias is detected, the results should be interpreted cautiously.

### Estimating the BPRF

Using the final linear mixed-effects model and the draws of RR with gamma, we estimated the BPRF[29]. The BPRF reflects the most conservative estimate of association between each of the exposures and the health outcome of interest that is consistent with the available evidence and can be compared across different risk–outcome pairs. For harmful risk factors such as childhood physical abuse, psychological abuse and neglect, the BPRF is calculated as the fifth quantile of draws closest to the null from the distribution defined by the RR UIs inclusive of between-study heterogeneity. From the BPRF, we derived the ROS as the signed natural log(BPRF) divided by two and a conservative estimate of the minimum increased risk of the health outcome due to exposure to the risk factor. The ROS reflects both the magnitude of the risk–outcome association and the consistency of findings studies informing the association. A large positive ROS nearing 1 indicates a strong association supported by consistent evidence, whereas a negative or small (<0.14) positive ROS reflects a weak association and/or a lack of consistent evidence of an association. ROS values can also be translated into star rating categories. The star ratings aim to aid in the interpretation and comparison of the ROS findings and range from one to five stars, adhering to previously published benchmarks set within

the BPRF methodology[29]. These thresholds were established to align with minimum excess risk values determined in consultation with clinical and methodological experts. For example, a one-star rating indicates weak evidence of association where there is a clear need for further research that may change the assessment of risk as the most conservative interpretation of existing data suggests the possibility of no excess risk. A two-star rating similarly indicates relatively weak evidence of an association with up to 15% increased risk. Additional stars reflect increasing evidence of an association up to a five-star rating, which suggests very strong evidence and a conservative estimate of an increase in risk by at least 85%. Zero-star risk–outcome pairs are not based on ROS values but are defined as pairs for which there is no indication of a statistically significant association between the risk and the health outcome (that is, the 95% UI without gamma crosses the null).

### Model validation

We undertook several additional sensitivity analyses to evaluate the robustness of our primary models' results to our data input, methodological approaches and model parameters. Owing to varied exposure definitions across studies and to assess the specificity of our findings to the perpetrator of violence, we executed two a priori sensitivity analyses in which we restricted our input data to observations examining violence perpetrated by unrestricted or unspecified perpetrators and as violence perpetrated by family or household members. Subsequently, we tested separate models for male- and female-specific observations. We additionally undertook several analyses in which we investigated the impact of selecting certain health outcome definitions for outcomes with various types of included outcome definitions (that is, outcome-specific sensitivity analyses). For example, the outcome-specific analyses included restricting data for major depressive disorder to only studies that explicitly state that major depressive disorder is their outcome of interest or to only studies that characterize it as the broader term of 'depression' as two distinct outcome-specific sensitivity analyses. For these models, the sole alteration in our model parameters was related to the implementation of the 10% data trimming, which is dependent on the number of observations available for each outcome model (that is, data are trimmed only if ten observations or more are included). Furthermore, for models in our primary analysis with ten or more observations, we ran an additional sensitivity analysis with the same data inputs and model parameters but without 10% trimming. Lastly, to test the impact of our approach of adjusting for multiple effect sizes from the same study, we also ran models with the original standard error associated with each data point. These analyses could be conducted only with a minimum of three eligible observations for a specified risk–outcome association. The number of studies and data points included in each sensitivity analysis are presented in Supplementary Tables 6–18.

### Statistical analysis and reproducibility

Analyses were carried out using R version 4.0.5 and Python version 3.10.9. This investigation relied on existing published data. No statistical method was used to predetermine sample sizes. For each health outcome, we included all studies that met our inclusion criteria. This study did not engage in primary data collection, randomization and blinding. Therefore, data exclusions were not relevant to the present study, and, as such, no data were excluded from the analyses. We have made our data and code available to foster reproducibility.

### Reporting summary

Further information on research design is available in the Nature Portfolio Reporting Summary linked to this article.

## Data availability

The findings from this study are supported by data extracted from published literature. Citations for all input data are provided as part

of this manuscript. Study characteristics and all included data points are provided in Supplementary Tables 3 and 34. Details on data sources can also be found via the GHDx website at https://ghdx.healthdata.org/record/ihme-data/health-effects-vac-bop-ros.

## Code availability

All code used for these analyses is publicly available via GitHub at https://github.com/ihmeuw-msca/burden-of-proof/. Analyses were carried out using R version 4.0.5 and Python version 3.10.9.

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

## Acknowledgements

Research reported in this publication was supported by the Bill & Melinda Gates Foundation (award number 66-7813, E.G.). The funders of the study had no role in the study design, data

collection, data analysis, data interpretation, writing of the final report or decision to publish.

## Author contributions

A.A., M.J.B.d.R., J.C., R.J.C.C.-A., S.C., J.K.C., C.V.N.C., F.M.D.d.A., G.N.d.A., G.F.G., B.H., M.H., M.K., R.Q.H.L., S.M., J.K.M., V.P., S.R., D.S., C.N.S., C.S., A.T., A.V. and N.V. were primarily responsible for seeking, cataloguing, extracting or cleaning data; and designing or coding figures and tables. A.A., J.C., R.J.C.C.-A., J.S.C., F.M.D.d.A., A.N.D., L.S.F., E.G., G.F.G., B.H., F.M.K., R.Q.H.L., N.M., V.P., D.S., C.N.S., C.S., H.S., A.V. and N.V. provided data or critical feedback on data sources. A.Y.A., F.B., R.J.C.C.-A., J.S.C., L.S.F., E.G., G.F.G., F.M.K., N.M., C.J.L.M., C.N.S., C.S., H.S. and P.Z. provided critical feedback on methods or results. F.B., J.S.C., E.D., L.S.F., E.G., G.F.G., S.I.H., F.M.K., S.A.M., C.J.L.M., N.M., C.N.S., C.S. and H.S. drafted the work or revised it critically for important intellectual content. L.S.F., E.G., E.C.M. and E.M.O. managed the overall research enterprise. A.Y.A., S.C., C.J.L.M., R.J.D.S. and P.Z. developed methods or computational machinery. L.S.F. and G.F.G. were primarily responsible for applying analytical methods to produce estimates. L.S.F., E.G. and G.F.G. wrote the first draft of the manuscript. L.S.F., E.G., S.I.H., E.C.M. and E.M.O. managed the estimation or publication process.

## Competing interests

C.S., L.S.F. and E.M. report support for the present manuscript from Bill & Melinda Gates Foundation. G.F.G. reports support for the present manuscript from Bill & Melinda Gates Foundation with payment through salary at IHME; Bloomberg Philanthropies with payment through salary at IHME. A.A. reports support for the present manuscript from Institute for Health Metrics and Evaluation. F.B. reports support for attending meetings and/or travel from Partnership for Maternal, Newborn and Child Health, Geneva, Switzerland and Fondation Botnar, Basel, Switzerland; leadership or fiduciary role in other board, society, committee or advocacy group, paid or unpaid as Chair of Governance and Ethics Committee for the Partnership of Maternal, Newborn and Child Health; International Advisory Board Chair of United Nations University International Institute for Global Health, Kuala Lumpur, Malaysia; Co-Chair of The Lancet Commission on Gender-Based Violence and the Maltreatment of Young People; Vice Chair Fondation Botnar, Basel, Switzerland; Member of The Lancet Future of Neonatology Commission; Member of The Lancet and Chatham House Commission on Universal Health Member of Lancet Commission on Investing in Health 3. J.K.C. reports grants or contracts from University of Warwick; support for attending meetings and/or travel from University of Miami, University of Washington. J.S.C. reports grants or contracts from National Institute for Health and Care Research, Youth Endowment Fund, College of Policing, University of Birmingham, Birmingham City Council; support for attending meetings and/or travel from University of Miami, University of Washington. B.H. reports grants or contracts from West Midlands Secure Data Environment (£50,000 Pump Priming Fund for project titled: Developing an automated evaluation pipeline to identify the effectiveness of digital interventions in acute care: A pilot study assessing inequalities in the effectiveness of DERM). F.M.K. reports grants or contracts from Merck KGaA/EMD Serono (research grant to University of Miami); Tides Foundation via the Oak Foundation (two research grants to the University of Miami); Fondation Botnar (research grant to the University of Miami); Finker-Frenkel Family Foundation (gift to the University of Miami to support the Lancet Commission on Gender-Based Violence and Maltreatment of Young People); Wellcome Trust (gift to the University of Miami to support the Lancet Commission on Gender-Based Violence and Maltreatment of Young People); Mena Catering (gift to the University of Miami to support the Lancet Commission on Gender-Based Violence and Maltreatment of Young People); Gloria Estefan Foundation (gift to the University of Miami to support the Lancet Commission on Gender-Based Violence and Maltreatment of Young People); Jose Milton Foundation (gift to the University of Miami to support the Lancet Commission on Gender-Based Violence and Maltreatment of Young People); reports consulting fees from Merck KGaA/EMD Serono (personal consulting agreement to advise company's research/dissemination strategy for 'Healthy Women, Healthy Economies' and 'Embracing Carers' initiative focused on caregiving and women in leadership. Totally unrelated to the subject of this paper/No child sexual abuse related work); Tecnológico de Monterrey (provide strategic guidance on research priorities and lectures for the Institute for Obesity Research at the Tecnólogico de Monterrey (university); totally unrelated to the subject of this paper/no child sexual abuse related work); leadership or fiduciary role in other board, society, committee or advocacy group, paid or unpaid from Founding President, Tómatelo a Pecho, A.C. (Mexican non-profit organization that has promoted research, advocacy, awareness and early detection of breast cancer since its inception and has since expanded to promote women's and girls health broadly and health systems); Esperanza United (Member, Board of Directors – unpaid); Senior Economist, Mexican Health Foundation (unpaid). S.R. reports support for the present manuscript from Vital Strategies and Bill and Melinda Gates Foundation. The other authors declare no competing interests.

## Additional information

**Extended data** is available for this paper at https://doi.org/10.1038/s41562-025-02143-3.

**Correspondence and requests for materials** should be addressed to Luisa S. Flor.

Luisa S. Flor[1,2] ✉, Caroline Stein [1,2], Gabriela F. Gil[1], Mariam Khalil[1], Molly Herbert[1], Aleksandr Y. Aravkin[1,3], Alejandra Arrieta[1,2], María Jose Baeza de Robba[4,5], Flavia Bustreo[6,7], Jack Cagney [1], Renzo J. C. Calderon-Anyosa[8], Sinclair Carr [1], Jaidev Kaur Chandan[9,10], Joht Singh Chandan [10], Carolina V. N. Coll[11,12], Fabiana Martins Dias de Andrade[13], Gisele N. de Andrade [13], Alexandra N. Debure [14], Erin DeGraw[1], Ben Hammond [10], Simon I. Hay [1,2], Felicia M. Knaul[15,16,17,18], Rachel Q. H. Lim[10], Susan A. McLaughlin[1], Nicholas Metheny [19], Sonica Minhas [10], Jasleen K. Mohr[10], Erin C. Mullany[1], Christopher J. L. Murray [1,2], Erin M. O'Connell[1], Vedavati Patwardhan[1,20], Sofia Reinach [21], Dalton Scott [14], Cory N. Spencer[1], Reed J. D. Sorensen[1], Heidi Stöckl [22], Aisha Twalibu[1,2], Aiganym Valikhanova[1], Nádia Vasconcelos[13], Peng Zheng[1,2] & Emmanuela Gakidou [1,2]

[1]Institute for Health Metrics and Evaluation, University of Washington, Seattle, WA, USA. [2]Department of Health Metrics Sciences, School of Medicine, University of Washington, Seattle, WA, USA. [3]Department of Applied Mathematics, University of Washington, Seattle, WA, USA. [4]School of Nursing, The Pontifical Catholic University of Chile, Santiago, Chile. [5]Center for Global Health Equity, University of Michigan, Ann Arbor, MI, USA. [6]Fondation Botnar, Basel, Switzerland. [7]Partnership for Maternal, Newborn and Child Health, Geneva, Switzerland. [8]McGill University, Montreal, Quebec, Canada. [9]Warwick Medical School, University of Warwick, Coventry, UK. [10]Institute of Applied Health Research, University of Birmingham, Birmingham, UK. [11]Department of Epidemiology, Federal University of Pelotas, Pelotas, Brazil. [12]Human Development and Violence Research Center, Federal University of Pelotas, Pelotas, Brazil. [13]Federal University of Minas Gerais, Belo Horizonte, Brazil. [14]School of Nursing and Health Studies, University of Miami, Coral Gables, FL, USA. [15]Institute for the Advanced Study of the Americas, University of Miami, Coral Gables, FL, USA. [16]Department of Medicine, David Geffen School of Medicine, UCLA, Los Angeles, CA, USA. [17]Escuela de Medicina y Ciencias de la Salud, Tecnológico de Monterrey Faculty of Excellence, Mexico City, Mexico. [18]Tómatelo a Pecho, A.C., Mexico City, Mexico. [19]Nell Hodgson Woodruff School of Nursing, Emory University, Atlanta, GA, USA. [20]Center on Gender Equity and Health, UC San Diego School of Medicine, San Diego, CA, USA. [21]Vital Strategies, New York, NY, USA. [22]Institute of Medical Information Processing, Biometry and Epidemiology, Ludwig-Maximilians-University Munich, Munich, Germany. ✉e-mail: lsflor@uw.edu

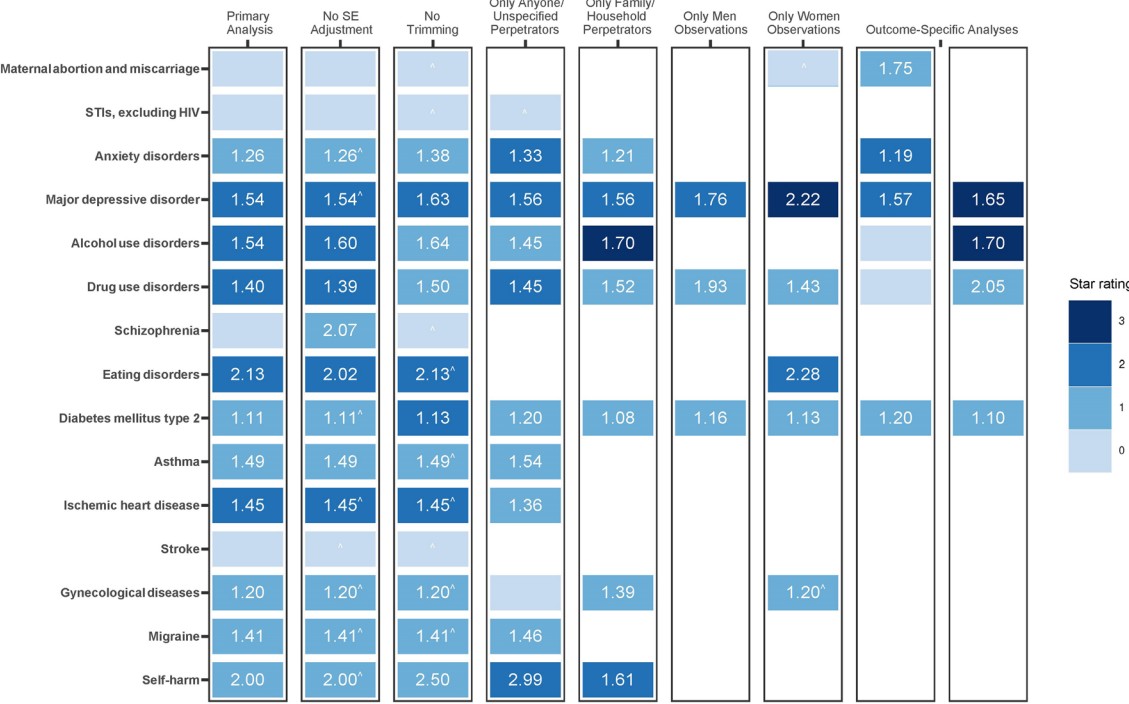

**Extended Data Fig. 1 | Mean relative risk and strength of evidence found in sensitivity analyses on the associations between childhood physical violence and various health outcomes.** This heatmap compares the estimated mean relative risk and the star rating for the association between childhood physical violence and each of the 13 health outcomes evaluated across different sensitivity analyses. The leftmost column, titled 'Primary analysis', reflects the primary results of the present manuscript. Each consecutive column reflects a different sensitivity analysis (described in more detail in the Methods) with specific model restrictions or restrictions to the input data, and if the cells are marked with a ˆ, they reflect a restriction that exactly matches the primary analysis. Each row corresponds to a different outcome of interest. The number overlayed on each cell corresponds to the mean relative risk estimate derived from each analysis for the relationship between the health outcome listed on the y-axis and childhood exposure to physical violence. Colored cells with no overlayed text reflect zero-star risk-outcome pairs in which the analysis in question did not find sufficient evidence of an association between the outcome and childhood physical violence. The cells are colored in accordance with the star rating based on a conservative interpretation of the data in the sensitivity analysis of interest on a 1 (weak evidence) to 5 (strong and consistent evidence of a strong association) star scale. Empty white cells reflect sensitivity analyses for which we did not have enough relevant data (three or more studies) to examine.

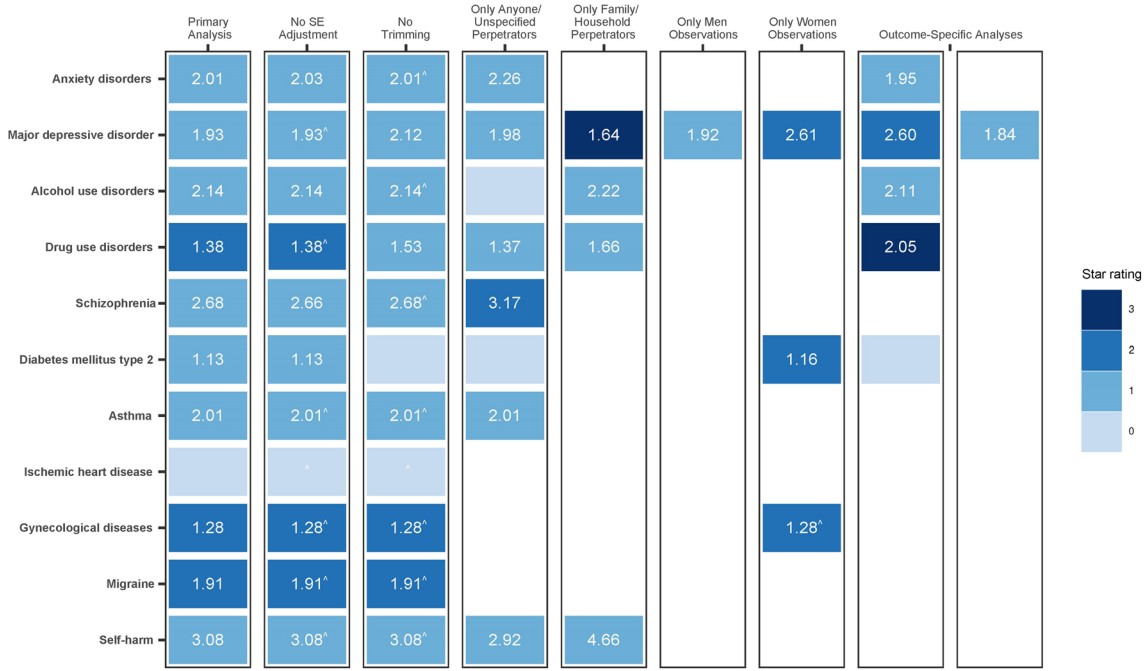

**Extended Data Fig. 2 | Mean relative risk and strength of evidence found in sensitivity analyses on the associations between childhood psychological violence and various health outcomes.** This heatmap compares the estimated mean relative risk and the star rating for the association between childhood psychological violence and each of the 10 health outcomes evaluated across different sensitivity analyses. The leftmost column, titled 'Primary analysis', reflects the primary results of the present manuscript. Each consecutive column reflects a different sensitivity analysis (described in more detail in the Methods) with specific model restrictions or restrictions to the input data, and if the cells are marked with a ^, they reflect a restriction that exactly matches the primary analysis. Each row corresponds to a different outcome of interest. The number overlayed on each cell corresponds to the mean relative risk estimate derived from each analysis for the relationship between the health outcome listed on the y-axis and childhood exposure to psychological violence. Colored cells with no overlayed text reflect zero-star risk-outcome pairs in which the analysis in question did not find sufficient evidence of an association between the outcome and childhood psychological violence. The cells are colored in accordance with the star rating based on a conservative interpretation of the data in the sensitivity analysis of interest on a 1 (weak evidence) to 5 (strong and consistent evidence of a strong association) star scale. Empty white cells reflect sensitivity analyses for which we did not have enough relevant data (three or more studies) to examine.

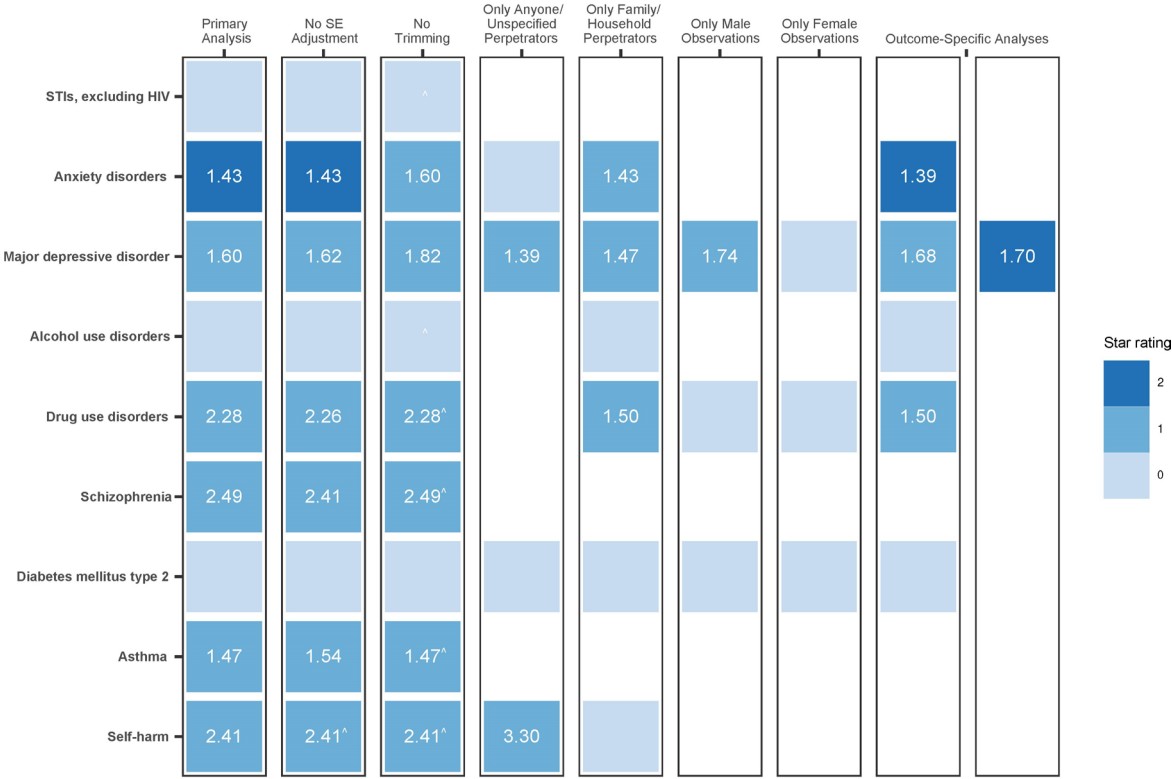

**Extended Data Fig. 3 | Mean relative risk and strength of evidence found in sensitivity analyses on the associations between childhood neglect and various health outcomes.** This heatmap compares the estimated mean relative risk and the star rating for the association between childhood neglect and each of the nine health outcomes evaluated across different sensitivity analyses. The leftmost column, titled 'Primary analysis', reflects the primary results of the present manuscript. Each consecutive column reflects a different sensitivity analysis (described in more detail in the Methods) with specific model restrictions or restrictions to the input data, and if the cells are marked with a ^, they reflect a restriction that exactly matches the primary analysis. Each row corresponds to a different outcome of interest. The number overlayed on each cell corresponds to the mean relative risk estimate derived from each analysis for the relationship between the health outcome listed on the y-axis and childhood exposure to neglect. Colored cells with no overlayed text reflect zero-star risk-outcome pairs in which the analysis in question did not find sufficient evidence of an association between the outcome and childhood neglect. The cells are colored in accordance with the star rating based on a conservative interpretation of the data in the sensitivity analysis of interest on a 1 (weak evidence) to 5 (strong and consistent evidence of a strong association) star scale. Empty white cells reflect sensitivity analyses for which we did not have enough relevant data (three or more studies) to examine.

# Reporting Summary

## Statistics

For all statistical analyses, confirm that the following items are present in the figure legend, table legend, main text, or Methods section.

| n/a | Confirmed | |
|---|---|---|
| ☐ | ☒ | The exact sample size (*n*) for each experimental group/condition, given as a discrete number and unit of measurement |
| ☐ | ☒ | A statement on whether measurements were taken from distinct samples or whether the same sample was measured repeatedly |
| ☐ | ☒ | The statistical test(s) used AND whether they are one- or two-sided *Only common tests should be described solely by name; describe more complex techniques in the Methods section.* |
| ☐ | ☒ | A description of all covariates tested |
| ☐ | ☒ | A description of any assumptions or corrections, such as tests of normality and adjustment for multiple comparisons |
| ☐ | ☒ | A full description of the statistical parameters including central tendency (e.g. means) or other basic estimates (e.g. regression coefficient) AND variation (e.g. standard deviation) or associated estimates of uncertainty (e.g. confidence intervals) |
| ☐ | ☒ | For null hypothesis testing, the test statistic (e.g. *F*, *t*, *r*) with confidence intervals, effect sizes, degrees of freedom and *P* value noted *Give P values as exact values whenever suitable.* |
| ☐ | ☒ | For Bayesian analysis, information on the choice of priors and Markov chain Monte Carlo settings |
| ☒ | ☐ | For hierarchical and complex designs, identification of the appropriate level for tests and full reporting of outcomes |
| ☐ | ☒ | Estimates of effect sizes (e.g. Cohen's *d*, Pearson's *r*), indicating how they were calculated |

*Our web collection on statistics for biologists contains articles on many of the points above.*

## Software and code

Policy information about availability of computer code

| Data collection | No primary data collection was carried out for this analysis. |
|---|---|
| Data analysis | All code used for these analyses is publicly available online (https://github.com/ihmeuw-msca/burden-of-proof/). |

For manuscripts utilizing custom algorithms or software that are central to the research but not yet described in published literature, software must be made available to editors and reviewers. We strongly encourage code deposition in a community repository (e.g. GitHub). See the Nature Portfolio guidelines for submitting code & software for further information.

## Data

Policy information about availability of data

All manuscripts must include a data availability statement. This statement should provide the following information, where applicable:
- Accession codes, unique identifiers, or web links for publicly available datasets
- A description of any restrictions on data availability
- For clinical datasets or third party data, please ensure that the statement adheres to our policy

The findings from this study are supported by data from the published literature. Study characteristics for all input data used in the analyses are provided in Supplementary Table 3. Data points included in each analysis are available in Supplementary Table 34. Input data and a lis of the included studies are also freely available at: https://ghdx.healthdata.org/record/ihme-data/health-effects-vac-bop-ros

## Human research participants

Policy information about studies involving human research participants and Sex and Gender in Research.

| | |
|---|---|
| Reporting on sex and gender | No primary data collection was carried out for this analysis, so the study does not involve human research participants. As stated in the methods overview, our estimates reflect both boys and girls, drawing upon all available data regardless of how or if the input study collected and reported data by sex or gender. |
| Population characteristics | No primary data collection was carried out for this analysis, so the study does not involve human research participants. The analysis evaluated the effect of childhood physical and psychological abuse and neglect (against both boys and girls) on selected disease endpoints. We accepted all studies regardless of the age of the study sample or target population. |
| Recruitment | No primary data collection was carried out for this analysis, so we did not recruit participants. |
| Ethics oversight | This study was approved by the University of Washington IRB Committee (study #9060). |

Note that full information on the approval of the study protocol must also be provided in the manuscript.

# Field-specific reporting

Please select the one below that is the best fit for your research. If you are not sure, read the appropriate sections before making your selection.

☒ Life sciences ☐ Behavioural & social sciences ☐ Ecological, evolutionary & environmental sciences

For a reference copy of the document with all sections, see nature.com/documents/nr-reporting-summary-flat.pdf

# Life sciences study design

All studies must disclose on these points even when the disclosure is negative.

| | |
|---|---|
| Sample size | The number of studies included was determined through a systematic literature review that included title/abstract screening, full-text screening, and citation searching to identify relevant articles and extract data points used as input to models. Details surrounding the sample size of each included study can be found in Supplementary Table 3 and the number of included studies per risk-outcome pair is reported in Table 2. |
| Data exclusions | As described in Supplementary Information Section 7.2, reports were excluded based on the following exclusion criteria:<br>Study design: Cross-sectional, ecological, case series or case studies.<br>Participants: Studies conducted in subgroups identified only by convenience sampling or subgroups identified via a shared characteristic that is likely related to risk of exposure to violence or the reported health outcome.<br>Exposure measurement: Studies that report only an aggregate measure of exposure combining exposure to a form of violence with other, non-eligible exposures (e.g., reports a composite ACE score only) were excluded. For these studies, we are unable to disentangle the effect of violence exposure from the effects of other hardships or exposure types, preventing their inclusion in our review.<br>Does not meet minimum reporting criteria: Studies missing essential data, that is, those that do not report effect sizes and uncertainty information (confidence intervals, sample sizes) or the data needed to impute an effect size with uncertainty information. |
| Replication | This is a meta-analysis of existing studies with many years of cohort and other data. The code and data used are publicly available, and the analyses could theoretically be replicated. |
| Randomization | This analysis is a meta-analysis of existing studies and thus, there were no experimental groups. |
| Blinding | N/A. This study was a meta-analyses using existing data and did not the collection of primary data nor experimental/control groups. As such, blinding was not relevant. |

# Reporting for specific materials, systems and methods

We require information from authors about some types of materials, experimental systems and methods used in many studies. Here, indicate whether each material, system or method listed is relevant to your study. If you are not sure if a list item applies to your research, read the appropriate section before selecting a response.

## Materials & experimental systems

| n/a | Involved in the study |
|-----|----------------------|
| ☒ | ☐ Antibodies |
| ☒ | ☐ Eukaryotic cell lines |
| ☒ | ☐ Palaeontology and archaeology |
| ☒ | ☐ Animals and other organisms |
| ☒ | ☐ Clinical data |
| ☒ | ☐ Dual use research of concern |

## Methods

| n/a | Involved in the study |
|-----|----------------------|
| ☒ | ☐ ChIP-seq |
| ☒ | ☐ Flow cytometry |
| ☒ | ☐ MRI-based neuroimaging |

