## [Peer Review File · Nature Human Behaviour]

Health effects associated with exposure of children to physical violence, psychological violence, and neglect: a Burden of Proof study

Corresponding Author: Professor Emmanuela Gakidou

Version 0:

Decision Letter:

19th August 2024

Dear Professor Gakidou,

Thank you once again for your manuscript, entitled "Health effects associated with exposure of children to physical violence, psychological violence, and neglect: a Burden of Proof study." Please accept my sincere apologies once again for the extraordinary delay in communicating this decision to you.

Your manuscript has now been evaluated by 3 reviewers, whose comments are included at the end of this letter. Although the reviewers find your work to be of interest, they also raise some important concerns. We are interested in the possibility of publishing your study in Nature Human Behaviour, but would like to consider your response to these concerns in the form of a revised manuscript before we make a decision on publication.

In revision, we ask that you pay particular attention to concerns raised by Reviewers 2 and 3, who point out that this is a saturated field and that the contribution of your work above and beyond existing evidence synthesis efforts is unclear. Please make sure to present the burden of proof methodology in sufficient detail (without relying on other publications), explaining any unique contribution of the approach. Additionally, please ensure that your search is comprehensive and includes all relevant studies (see Reviewer 3's relevant comment). Finally, please address concerns related to polyvictimization and how it affects your estimates.

Your revised manuscript must comply fully with our editorial policies and formatting requirements. Failure to do so will result in your manuscript being returned to you, which will delay its consideration. To assist you in this process, I have attached a checklist that lists all of our requirements. If you have any questions about any of our policies or formatting, please don't hesitate to contact me.

In sum, we invite you to revise your manuscript taking into account all reviewer and editor comments. We are committed to providing a fair and constructive peer-review process. Do not hesitate to contact us if there are specific requests from the reviewers that you believe are technically impossible or unlikely to yield a meaningful outcome.

We hope to receive your revised manuscript within four months. I would be grateful if you could contact us as soon as possible if you foresee difficulties with meeting this target resubmission date.

- Include a "Response to the editors and reviewers" document detailing, point-by-point, how you addressed each editor and referee comment. If no action was taken to address a point, you must provide a compelling argument. When formatting this document, please respond to each reviewer comment individually, including the full text of the reviewer comment verbatim followed by your response to the individual point. This response will be used by the editors to evaluate your revision and sent back to the reviewers along with the revised manuscript.
- Highlight all changes made to your manuscript or provide us with a version that tracks changes.

Link Redacted

We look forward to seeing the revised manuscript and thank you for the opportunity to review your work. Please do not hesitate to

contact me if you have any questions or would like to discuss these revisions further.

Sincerely,

[REDACTED]

[REDACTED] PhD

Nature Human Behaviour

Reviewer expertise:

Reviewer #1: child abuse/maltreatment; systematic review and meta-analysis

Reviewer #2: childhood adversity/trauma

Reviewer #3: child trauma/maltreatment; systematic review and meta-analysis

REVIEWER COMMENTS:

Reviewer #1:

Remarks to the Author:

This study presents a meta-analysis supplemented by burden-of-proof methodology to evaluate the association between childhood exposure to physical and psychological violence and neglect with a range of health outcomes. This is a very thorough and well-conducted study and I have only a few suggestions for improvement.

There is some mention in the discussion of polyvictimization, the tendency for multiple forms of childhood maltreatment (sexual abuse, physical violence, psychological violence, and neglect) to commonly co-occur. However, it is simply stated that the current study could not account for this in the analysis, a methodological limitation of the analysis. I believe the important of polyvictimization warrants brief mention. Unfortunately, deleterious effects of multiple forms of childhood maltreatment can compound with overlapping exposures. An implication of this is that many of the effects (e.g., an at least 19% increase risk for major depression with physical violence exposure) may in reality be higher when accounting for overlap with other forms of maltreatment that may also contribute to risk for the same outcome.

Finkelhor, D., Ormrod, R. K., & Turner, H. A. (2007). Poly-victimization: A neglected component in child victimization. *Child Abuse & Neglect*, 31, 7–26.

Finkelhor, D., Ormrod, R. K., & Turner, H. A. (2009). Lifetime assessment of poly-victimization in a national sample of children and youth. *Child Abuse & Neglect*, 33, 403–411.

In the limitations section, it was mentioned that it was not possible to disaggregate effects of violence by age of exposure. In addition to that, it could be mentioned that it was not possible to control for the timing of assessment of exposures and assessment of outcomes. The degree of risk may differ as a function of time in between. Related to this, there is evidence that retrospective versus prospective studies may yield different findings. For example, retrospective studies appear to yield strong associations with depression than prospective studies.

Colman, I., Kingsbury, M., Garad, Y., Zeng, Y., Naicker, K., Patten, S., Jones, P. B., Wild, T. C., & Thompson, A. H. (2016). Consistency in adult reporting of adverse childhood experiences. *Psychological Medicine*, 46, 543–549.

Patten, S. B., Wilkes, T. C. R., Williams, J. V. A., Lavorato, D. H., El-Guebaly, N., Schopflocher, D., Wild, C., Colman, I., & Bulloch, A. G. M. (2015). Retrospective and prospectively assessed childhood adversity in association with major depression, alcohol consumption and painful conditions. *Epidemiology and Psychiatric Sciences*, 24, 158–165.

Reviewer #2:

Remarks to the Author:

This study reports a systematic review and meta-analysis examining associations between childhood physical violence, psychological threat, and neglect with a variety of health outcomes.

While an impressive undertaking in terms of the sheer amount of work conducted, there were several issues that tempered my enthusiasm for publication.

My major issues with the work are as follows:

I am not sure it will be readily interpretable by researchers in my discipline, particularly how it extends the existing literature. There is already a wealth of literature documenting associations between childhood maltreatment and adverse health outcomes. In the field, this is being extended upon by examining the causal effect that removes confounding genetic or environmental factors, or looking at the mechanisms through which childhood adversity is linked to subsequent health outcomes. While I appreciate this study employs novel methodology, it does not make clear how this improves upon our existing knowledge, which has established the link between childhood violence and adult health outcomes. Clearly elucidating this would help evaluate its potential contribution to the literature.

Another major flaw is the fact that physical and psychological abuse, and neglect, show substantial co-occurrence within the individual i.e. children exposed to all types. This is briefly mentioned in the discussion, but it is not made clear in the introduction or methods what the rationale was for examining types of childhood maltreatment separately, given there is such high co-occurrence, and therefore very difficult to attribute health outcomes to one type of maltreatment when other types have not been controlled for or examined.

It was also difficult to understand not only the burden of proof methodology itself but also how this is an improvement on other methods. At present the manuscript cannot stand on its own in terms of understanding the methodology as the reader has to seek out additional references to understand this.

This manuscript could be improved through more succinct reporting in all sections. At present the length and amount of information are prohibitive in understanding this work and its implications.

More minor suggestions are included below:

Abstract

The term "violence against children" doesn't seem to be the best term for the exposure, given this study also included neglect and psychological forms of maltreatment. I would suggest another term that encompasses all the exposures better

Methods

It was not clear to me how the star rating was determined. There is a reference to previously published benchmarks but this should be briefly summarised in the main text of this manuscript

It would be helpful to include a table listing the outcomes that were searched for, and found studies for. Also specify what outcomes are included in the term "gynecological diseases"

It is also important to acknowledge potential genetic and environmental confounding in the relationship between childhood violence and outcomes e.g. schizophrenia. I direct the authors to literature investigating this issue:

Baldwin, J. R., Sallis, H. M., Schoeler, T., Taylor, M. J., Kwong, A. S. F., Tielbeek, J. J., Barkhuizen, W., Warriar, V., Howe, L. D., Danese, A., McCrory, E., Rijdsdijk, F., Larsson, H., Lundström, S., Karlsson, R., Lichtenstein, P., Munafò, M., & Pingault, J. B. (2023). A genetically informed Registered Report on adverse childhood experiences and mental health. *Nat Hum Behav*, 7(2), 269-290. <https://doi.org/10.1038/s41562-022-01482-9>

Baldwin, J. R., Wang, B., Karwatowska, L., Schoeler, T., Tsaligopoulou, A., Munafò, M. R., & Pingault, J.-B. (2023). Childhood Maltreatment and Mental Health Problems: A Systematic Review and Meta-Analysis of Quasi-Experimental Studies. *American Journal of Psychiatry*, appi.ajp.20220174. <https://doi.org/10.1176/appi.ajp.20220174>

Why were 3 studies chosen as the minimum? The rationale should at least be included

It is mentioned that a protocol was published – whether it was prospectively published prior to beginning the systematic review should be included.

Results

Page 7 line 147: suggest including the possible range for ROS to aid interpretation of a score of 0.25

Page 7 lines 147-150 seem better suited to the methods section

Discussion

The findings of this study contrast against a wealth of literature that report strong associations between childhood violence and many of the health outcomes reviewed. Therefore, the discussion should include greater attention to why the results of this review were so different.

Table 1 – I'm not sure why policy implications are included as a table and not in text in the discussion.

Reviewer #3:

Remarks to the Author:

Thank you for the opportunity to review this paper. It presents a novel analytical framework to summarise the evidence of associations between violence against children (aka maltreatment) and health outcomes. The paper employs an interesting analytical approach building on previously published work already including child sexual abuse. I have some comments, which I hope may be helpful is revising the manuscript.

COMMENTS

There is extensive evidence synthesis work on the links between violence against children (aka childhood maltreatment) and various health outcomes. The literature is particularly saturated for the associations with mental health outcomes, which is extensively discussed here, too. The authors should clarify the novelty of their work in the context of the literature, which they consider to a minimal extent. In particular, I appreciate they use a novel metric, but I am unclear whether this offers advantages over previous work (please see other comments below).

The systematic review methodology is overall solid. However, I note that the analyses do not seem to include important studies in the area (e.g., Dunedin Study, E-Risk Study), which raises concerns about the comprehensiveness of the data included in the analyses.

Authors should clarify if there were any studies omitted for lack of information in the original papers / after requesting information from the authors.

On page 24, can the authors explain how they ensure that 'the most adjusted effect size' was always the best choice for data selection? For example, would 'the most adjusted effect size' be the one that adjusts for the co-occurrence of other maltreatment types? Or genetic influences on the health outcomes considered? These are important choices with considerable implications for the interpretation of the results.

The authors did not seem to include any information on the source of maltreatment, which has proven to be an important characteristic in modifying associations with health outcomes. For example, prospective and retrospective measures identify different people (<https://jamanetwork.com/journals/jamapsychiatry/fullarticle/2728182>) and are differentially associated with psychopathology (<https://jamanetwork.com/journals/jamapsychiatry/article-abstract/2818046>). As such, ignoring these differences may lead to measurement error, which would dilute the associations found. It is also a critical point for interpretation of the findings: while prevention of the experiences is necessary, this is unlikely in itself to address the burden of illness discussed (which would likely additionally require addressing the individual experiences).

In the second paragraph on page 26 (section on 'Testing and adjusting for biases across study designs and characteristics'), the algorithm / statistical approach needs to be clarified for the readers.

Authors should provide a clearer and more accessible description of their methods involving the proof of risk function and model validation for non-specialist (pages 27-28), possibly referring to further supplementary materials. This text presents the innovation in the paper, and very few readers will be able to understand it and evaluate it. Beyond the steps taken, authors should provide clearer rationale for each of the step so as to walk the readers through their work.

The estimates presented look comparatively small and weak beyond a few examples. In contrast, the authors conclude that policymakers and health professionals should prioritise the elimination of all forms of violence against children (which, of course, should be a priority even at a moral level and has been highlighted many times previously). It would be helpful if authors could better link their results and conclusions.

The authors looked at associations with different types of childhood maltreatment but used a different label here (violence against children). I don't think that the label works because the experience includes neglect, which is not by definition a form of violence. I appreciate this may be a terminology suggested by UN, but it will not be familiar to most clinicians and public health practitioners. The use of different terms may also add confusion to an already fragmented literature.

Version 1:

Decision Letter:

Our ref: NATHUMBEHAV-24020706A

27th December 2024

Dear Dr. Gakidou,

Thank you for submitting your revised manuscript "Health effects associated with exposure of children to physical violence, psychological violence, and neglect: a Burden of Proof study" (NATHUMBEHAV-24020706A). It has now been seen by the original referees and their comments are below.

As you can see, Reviewers 1 and 2 find that the paper has improved in revision. Reviewer 3, however, remains unpersuaded about the value of the burden of proof methodology and the extent to which the insights arising from the current analyses reflect the complexity of health effects linked to violence and neglect. We appreciate the continuing concerns this reviewer expresses, which are likely to be shared by other researchers in the field. At the same time, we believe that at this point it should be up to our readers to determine the value and contribution of the work. We can therefore offer in principle to publish your article, provided that you make a final effort to address the comments raised by Reviewer 3 (as well as Reviewer 2), including by a more in-depth discussion of limitations and caveats on interpretation.

We are now performing detailed checks on your paper and will send you a checklist detailing our editorial and formatting requirements within two weeks. Please do not upload the final materials and make any revisions until you receive this additional information from us.

Sincerely,

[REDACTED]

[REDACTED] PhD

[REDACTED]
Nature Human Behaviour

Reviewer #1 (Remarks to the Author):

I believe the authors have been thoroughly responsive to reviewers comments. I appreciate their edits to increase accessibility to the unique add of their analytical approach to the literature. I have no additional comments for further revision.

Reviewer #2 (Remarks to the Author):

The reviewers have greatly strengthened the manuscript by highlighting the advantages of the burden of proof methodology. They have satisfactorily addressed most of my comments, yet two minor points remain.

I and other reviewers commented on the use of the term "violence against children" when it includes exposures like neglect. I respect that the authors wish to retain the term "violence against children", but their response doesn't adequately address the issue of describing neglect as an act of violence. The paragraph they have included in response should include mention of neglect specifically, perhaps at the end of the second sentence, or the final sentence.

Lines 496-501: "The ICVAC guidance does not distinguish between violence against children and childhood maltreatment but, rather, suggests that these terms are interchangeable and synonymous. Within this framework, childhood exposure to violence is categorized as any exposure occurring before the age of 18. The present analyses similarly use this demarcation line of 18 years old and adopt the terminology presented in the ICVAC. The ICVAC classifies VAC based on the nature of violent acts, defining each act as either the commission or omission of intentional behavior."

The second point is to do with acknowledging the potential for genetic and environmental confounding that I raised previously (Reviewer 2, point 8). The authors perhaps misunderstood my point, which was that children with a parent suffering a mental illness or substance use disorder may be more likely to be the victim of violence, but also more likely to develop a mental disorder or other outcome studied due to genetic liability. Therefore, without rigorous assessment, we cannot rule out these potential sources of confounding and say that violence accounts for all of this association when it may be that this is explained in part by the genetic component. So my point was more about mentioning in the discussion that these associations do not reflect causality and controlling for important genetic and environmental confounds would likely temper the magnitude of these associations. I still think this is important to mention briefly in the discussion.

Otherwise I am satisfied with the authors' response.

Reviewer #3 (Remarks to the Author):

In the revised manuscript, the authors failed to explain convincingly the novelty of the statistical methods and the importance of the findings in a language that is accessible to an audience of my peers. Furthermore, despite the statistical advances proposed, the methods still cannot address key concerns in the field including informant effects or genetic confounding. As such, the overall innovation appears limited in the context of the existing literature.

To expand on one of these aspects, in response to my comments as well as comments by other reviewers, the author did not address the point that, by their nature, the victimisation experiences that they assess as separate are very often interdependent. The authors now explain that the selected, where possible, samples with exposure to individual maltreatment types (which are not representative of victimisation occurrence in the population); alternatively, they selected effect sizes in analyses adjusting for co-occurrence (which artificially reduces the strength of associations observed modelling poly-victimisation given the natural co-occurrence). As such, this choice is misguided and offers a very partial overview of the field. It also likely contributes explaining why results somewhat differ from the findings of other published in other systematic work.

If the authors still want to use primarily the term 'violence against children', they should also include the term child maltreatment as key term and ideally in the abstract to ensure that the study is indexed more broadly.

Version 2:

Decision Letter:

Dear Emmanuela,

I am very pleased to inform you that your Article "Health effects associated with exposure of children to physical violence, psychological violence, and neglect: a Burden of Proof study", has now been accepted for publication in Nature Human Behaviour.

With warm regards,

Stavroula

 PhD
Nature Human Behaviour

P.S. Click on the following link if you would like to recommend Nature Human Behaviour to your librarian
<http://www.nature.com/subscriptions/recommend.html#forms>

** Visit the Springer Nature Editorial and Publishing website at http://editorial-jobs.springernature.com?utm_source=ejp_NHumB_email&utm_medium=ejp_NHumB_email&utm_campaign=ejp_NHumB for more information about our career opportunities. If you have any questions please click [here](mailto:editorial.publishing.jobs@springernature.com).

Response to Reviewers for NATHUMBEHAV-24020706:

Health effects associated with exposure of children to physical violence, psychological violence, and neglect: a Burden of Proof study

REVIEWER COMMENTS:

Reviewer #1:

This study presents a meta-analysis supplemented by burden-of-proof methodology to evaluate the association between childhood exposure to physical and psychological violence and neglect with a range of health outcomes. This is a very thorough and well-conducted study and I have only a few suggestions for improvement.

We appreciate the reviewer's kind words and insightful suggestions for our paper, which we believe have strengthened our manuscript. Please find our responses to your comments below.

1. There is some mention in the discussion of polyvictimization, the tendency for multiple forms of childhood maltreatment (sexual abuse, physical violence, psychological violence, and neglect) to commonly co-occur. However, it is simply stated that the current study could not account for this in the analysis, a methodological limitation of the analysis. I believe the importance of polyvictimization warrants brief mention. Unfortunately, deleterious effects of multiple forms of childhood maltreatment can compound with overlapping exposures. An implication of this is that many of the effects (e.g., an at least 19% increase risk for major depression with physical violence exposure) may in reality be higher when accounting for overlap with other forms of maltreatment that may also contribute to risk for the same outcome.

- Finkelhor, D., Ormrod, R. K., & Turner, H. A. (2007). *Poly-victimization: A neglected component in child victimization*. *Child Abuse & Neglect*, 31, 7–26.
- Finkelhor, D., Ormrod, R. K., & Turner, H. A. (2009). *Lifetime assessment of poly-victimization in a national sample of children and youth*. *Child Abuse & Neglect*, 33, 403–411.

Thank you for your thoughtful comment. In the revised manuscript, we have highlighted the importance of polyvictimization as well as the limitations of the available data to properly study it and account for it, including the potential implications of our findings in the context of poly-victimization (Discussion lines 382-393):

Lines 382-393: "Additionally, our analysis focused on reports of distinct forms of child abuse, and it did not account for the co-occurrence of violence types that often take place together^{172,173}. Co-occurrence of different forms of violence may act as compounding risk factors¹⁷⁴, particularly when considering the outcomes we found to be shared across physical and psychological violence and neglect, like major depressive disorder. Unfortunately, we found that most studies did not explicitly state whether their case definitions were exclusive to the type of violence in question, if they also included children who had concurrently experienced other forms of violence, or if they had even investigated the possibility of poly-victimization. On occasion, a study provided multiple data points accounting for different forms of violence. In these cases, we selected the data points that were explicitly restricted to individuals who had only experienced the violence type in question or that controlled for exposure to another form of

violence. However, there was insufficient data on specific combinations of violence to fully explore the consequences of co-occurrence.”

We described in more detail why we were unable to fully account for poly-victimization in our analysis – the fact that most studies we identified did not explicitly account for the co-occurrence of violence – and brought this issue to the forefront of our call to expand the scope of existing research to include compounding risks. We thank the reviewer for providing useful references, which we incorporated as references 173 and 174 in this section.

2. In the limitations section, it was mentioned that it was not possible to disaggregate effects of violence by age of exposure. In addition to that, it could be mentioned that it was not possible to control for the timing of assessment of exposures and assessment of outcomes. The degree of risk may differ as a function of time in between. Related to this, there is evidence that retrospective versus prospective studies may yield different findings. For example, retrospective studies appear to yield strong associations with depression than prospective studies.

- *Colman, I., Kingsbury, M., Garad, Y., Zeng, Y., Naicker, K., Patten, S., Jones, P. B., Wild, T. C., & Thompson, A. H. (2016). Consistency in adult reporting of adverse childhood experiences. *Psychological Medicine*, 46, 543–549.*
- *Patten, S. B., Wilkes, T. C. R., Williams, J. V. A., Lavorato, D. H., El-Guebaly, N., Schopflocher, D., Wild, C., Colman, I., & Bulloch, A. G. M. (2015). Retrospective and prospectively assessed childhood adversity in association with major depression, alcohol consumption and painful conditions. *Epidemiology and Psychiatric Sciences*, 24, 158–165.*

Thank you for bringing up these important points. We have added the important limitations of potential recall issues with retrospective reports to the Discussion section (lines 357-365) with reference to the first article suggested by the reviewer and another new reference on differences between different forms of reporting (references 161 and 162):

Lines 357-365: “There are also substantial challenges when measuring these characteristics, including potential recall bias or dissociative amnesia when self-reporting exposure and the under-reporting of violence through formal channels, which can affect the magnitude of the associations identified^{161,162}.[...] However, our ability to control for other dimensions of violence exposure and assessment was constrained by the dearth of data and, in some cases, by the absence of a clear gold-standard practice for exposure assessment.”

While the range of study designs included in our analyses makes it impractical to methodologically control for the lag between the assessment of exposures and assessment of outcomes, we ran an exploratory comparison between the relative risks reported and the follow-up time for the cohort studies where this information is available, and we did not find any clear patterns over follow-up time.

As to the reviewer’s second point regarding comparing retrospective reports of violence to prospective reports, we agree that this is important and added language reflecting this in our Discussion (lines 363-367):

Lines 363-367: “For example, both prospective (in the form of formal reports) and retrospective assessments of childhood violence have substantial weaknesses that may result in under-reporting or skewed reporting. Consequently, this limitation may have contributed to the

unexplained between-study heterogeneity estimated as part of the Burden of Proof methodology.”

However, there is ambiguity regarding the direction of the potential bias in our current analysis because many of the prospective reports we identified for inclusion rely on instances of violence against children that are reported through formal channels. As such, the exposure to violence against children may be under-reported in these studies. In contrast, retrospective reports are generally self-reported, which may introduce recall or other reporting biases. For example, more time between incidence and being asked about exposure may result in an individual foregoing reporting or inaccurately reporting the nature of an instance of violence in their childhood. On the other hand, there may also be lapses in self-reporting closer to the instance of violence due to trauma responses and dissociative amnesia. The absence of a clear gold-standard to designate as the reference value for a putative bias covariate makes direct adjustment/incorporation into our models challenging, but we believe this is an important area for future research.

Reviewer #2:

This study reports a systematic review and meta-analysis examining associations between childhood physical violence, psychological threat, and neglect with a variety of health outcomes. While an impressive undertaking in terms of the sheer amount of work conducted, there were several issues that tempered my enthusiasm for publication.

We appreciate the reviewer’s time and careful attention to detail in reviewing our manuscript. We hope the revisions we have made (described below) adequately address the reviewer’s concerns, and we believe these changes have improved the presentation of what we anticipate to be an important contribution to the field.

1. I am not sure it will be readily interpretable by researchers in my discipline, particularly how it extends the existing literature. There is already a wealth of literature documenting associations between childhood maltreatment and adverse health outcomes. In the field, this is being extended upon by examining the causal effect that removes confounding genetic or environmental factors, or looking at the mechanisms through which childhood adversity is linked to subsequent health outcomes. While I appreciate this study employs novel methodology, it does not make clear how this improves upon our existing knowledge, which has established the link between childhood violence and adult health outcomes. Clearly elucidating this would help evaluate its potential contribution to the literature.

We thank the reviewer for mentioning that the original manuscript did not clearly communicate the value added of our study over the existing literature. In response to this comment, we have revised the manuscript to better communicate the novelty of our findings and methodology in the context of existing research. We have added more detail regarding the advantages of the Burden of Proof methodology early in the manuscript, in the Introduction (lines 109-115) and the Methods (lines 432-448):

Lines 109-115: “This novel analytical framework incorporates—among other systematic modeling components—covariate selection and adjustment to account for known variation in input study characteristics and further quantifies remaining ‘unexplained’ between-study heterogeneity using random effects modeling (e.g., gamma and its uncertainty). Therefore, this

method more fully accounts for heterogeneity compared to traditional meta-analytic approaches, providing uniquely robust, credible, and well-specified evidence for policymakers and public health professionals dedicated to reducing VAC and its impacts.”

Lines 432-448: “This methodology utilizes a meta-regression tool—Bayesian, regularized, trimmed (MR-BRT)—to derive pooled relative risk estimates and associated uncertainty. These methods improve estimation accuracy by detecting and trimming potentially distorting outliers in the input data using a robust, likelihood-based approach and by rigorously testing and adjusting for bias covariates that capture systematic variability across known input study characteristics. Additionally, they quantify remaining heterogeneity across input studies that is otherwise not explained, which is incorporated into uncertainty estimates serving as the basis for generating a Burden of Proof Risk Function (BPRF), Risk Outcome Score (ROS), and star ratings – all reflective of both the strength of the association and the strength of the underlying evidence. These measures of evidence strength generated by the Burden of Proof approach provide a crucial complement to existing approaches that evaluate strength of evidence based on expert judgement¹⁷⁶. The methods have already been employed to evaluate health impacts linked with childhood sexual violence and intimate partner violence¹⁰, active¹⁷⁷ and passive smoking¹⁷⁸, chewing tobacco¹⁷⁹, high systolic blood pressure¹⁸⁰, and intake of unprocessed red meat¹⁸¹ and vegetables¹⁸². Our analysis further expands the Burden of Proof literature to include additional forms of violence against children, producing estimates using the same metrics and thus directly comparable to prior Burden of Proof findings, establishing violence against children as a health risk factor akin to smoking and high systolic blood pressure.”

Our manuscript synthesizes all available literature examining the health effects of physical abuse, psychological abuse, and neglect of children spanning seven databases and over 50 years, and, as the reviewer mentioned, affirmed the link between childhood violence and adverse health outcomes. The vast scope of our underlying systematic review and our implementation of Burden of Proof methods that allow us to rigorously evaluate the strength of existing evidence underlying these associations are two ways in which our study goes beyond the existing literature and makes a meaningful contribution. We further examine our findings in context of prior literature in the Discussion (lines 285-290; 292-293; 299-308):

Lines 285-290: “Particularly given our conservative interpretation of the evidence, the adverse health consequences found by our analysis in association with violence against children—which had already been underscored by some traditional meta-analytical efforts^{141,142}—reinforces the need for public health approaches to encourage the prevention of all forms of childhood violence, to better address the health needs of survivors, and to bolster the evidence base relating health consequences to childhood violence exposure.”

Lines 292-293; 299-308: “Consistent with the literature, we found that all forms of violence were associated with significantly greater risks for major depressive disorder and anxiety disorders^{9,143,144}. [...] Notably, in our analysis, we adhered to the GBD definitions of depression and anxiety, ensuring that only studies with rigorous mental health diagnosis definitions were included. This approach represents an improvement over previous scientific efforts that captured self-reported depressive or anxious symptoms, without diagnosis, as part of their outcome definition^{25,146}.

Recent meta-analytical efforts have also showcased substance use disorders as another important category of increased health risks, with age-, gender- and violence type-dependent associations to childhood violence exposure^{17,147,148}.”

In contrast to most prior systematic reviews, which largely focus on adverse childhood experiences broadly or a single sub-type of VAC, our review sets out to disentangle and compare the health consequences of three different forms of VAC, providing policy-relevant insights into forms of VAC that require distinct interventions. Moreover, our review was not restricted to specific health outcomes, unlike most previous reviews that narrowly focus primarily on some subset of reproductive outcomes, mental health, or substance use disorders. This broad approach allowed us to capture the wide breadth of health consequences of VAC without preconceptions of what health outcomes may be the most relevant, while drawing on the GBD framework ensured that our outcome definitions were rigorous and clinically relevant. Lastly, our findings provide robust and highly specified details about the relationships between childhood violence and health outcomes in a way that allow for comparisons with other risk factors in studies like the Global Burden of Disease study. This is important because what is lacking in the field is a way to compare exposure to childhood violence to other risk factors, such as obesity, stunting and wasting, air pollution and thus putting exposure to violence on the same scale as other priorities and highlighting its magnitude.

2. Another major flaw is the fact that physical and psychological abuse, and neglect, show substantial co-occurrence within the individual i.e. children exposed to all types. This is briefly mentioned in the discussion, but it is not made clear in the introduction or methods what the rationale was for examining types of childhood maltreatment separately, given there is such high co-occurrence, and therefore very difficult to attribute health outcomes to one type of maltreatment when other types have not been controlled for or examined.

We appreciate the reviewer highlighting this complex issue of co-occurrence, and we have expanded the Discussion to add more details on the rational and implications of our approach (lines 382-393). We have also extended the Methods section to explain our decision to establish separate case definitions for each type of violence (lines 505-508; 530-533) and to elaborate on the steps we took when feasible to mitigate the impact of potential co-occurrence on the results (lines 546-551):

Lines 382-393: “Additionally, our analysis focused on reports of distinct forms of child abuse, and it did not account for the co-occurrence of violence types that often take place together^{172,173}. Co-occurrence of different forms of violence may act as compounding risk factors¹⁷⁴, particularly when considering the outcomes we found to be shared across physical and psychological violence and neglect, like major depressive disorder. Unfortunately, we found that most studies did not explicitly state whether their case definitions were exclusive to the type of violence in question, if they also included children who had concurrently experienced other forms of violence, or if they had even investigated the possibility of poly-victimization. On occasion, a study provided multiple data points accounting for different forms of violence. In these cases, we selected the data points that were explicitly restricted to individuals who had only experienced the violence type in question or that controlled for exposure to another form of violence. However, there was insufficient data on specific combinations of violence to fully explore the consequences of co-occurrence.”

Lines 505-508: “In adherence to the ICVAC classification and given limitations in the data explored below, we have elected to examine the different forms of violence independently, based on the assumption that while a single individual may often be the survivor of multiple forms of VAC, the health implications may be distinct.”

Lines 530-533: “While we explored the feasibility of including combined forms of violence in our analysis, there were insufficient studies with comparable combined case definitions to draw reasonable conclusions regarding the patterns of compounding risk.”

Lines 546-551: “First, to reduce the potential impact of co-occurrent forms of violence, we prioritized analytical samples where the exposed groups were limited to individuals who had only experienced the violence type of interest. If information regarding restrictions on the case definitions was not available or no restriction was made on the basis of other exposures to violence, we selected effect sizes that were controlled for alternative exposures to violence over the effect sizes from the same study from samples that were not explicitly restricted or adjusted for alternative exposures.”

In brief, our analysis adopts the International Classification of Violence Against Children^z as its definitional framework, which presents distinct categories of violence, and which is the only classification scheme that has been adopted and approved at the global level. We believe it is important to provide estimates of the health effects associated with each of the sections of the ICVAC as this framework is being used by UNICEF, WHO, and many multilateral and international organizations that work in the field of violence against children.

We also explored the feasibility of incorporating information about the co-occurrence of violence, but data constraints rendered it challenging to systematically account for co-occurrence. We were constrained by the fact that most studies in our analyses do not explicitly account for co-occurrence, and among those that did, there were inconsistencies across combined case definitions and study designs. That said, in our analysis, when possible, we did prioritize data points that controlled for multiple exposures. The fact that data limitations restricted our ability to examine co-occurring violence in our analysis keenly highlights a need for expanded research and data on compounding violence against children.

^z UNICEF. International Classification of Violence against Children (ICVAC). *UNICEF DATA* <https://data.unicef.org/resources/international-classification-of-violence-against-children/> (2024).

3. It was also difficult to understand not only the burden of proof methodology itself but also how this is an improvement on other methods. At present the manuscript cannot stand on its own in terms of understanding the methodology as the reader has to seek out additional references to understand this.

We thank the reviewer for highlighting this. While the Burden of Proof methodology has been extensively peer-reviewed to date, it remains fairly new and so we appreciate the need for more information to be provided in our manuscript. In response to the Reviewer’s comment, we have both substantially revised the language used to include additional information on the Burden of Proof methodology throughout the Methods section (lines 575-655) and more explicitly described the strengths of this methodology in both the Introduction (lines 109-115) and the Methods (lines 432-448). While we are restricted in our ability to provide comprehensive details due to word limits and therefore must still, in some instances, point to previously published

materials, we hope our revisions provide sufficient detail to make the methods easier to understand without needing external references.

Lines 109-115: “This novel analytical framework incorporates—among other systematic modeling components—covariate selection and adjustment to account for known variation in input study characteristics and further quantifies remaining ‘unexplained’ between-study heterogeneity using random effects modeling (e.g., gamma and its uncertainty). Therefore, this method more fully accounts for heterogeneity compared to traditional meta-analytic approaches, providing uniquely robust, credible, and well-specified evidence for policymakers and public health professionals dedicated to reducing VAC and its impacts.”

Lines 432-448: “This methodology utilizes a meta-regression tool—Bayesian, regularized, trimmed (MR-BRT)—to derive pooled relative risk estimates and associated uncertainty. These methods improve estimation accuracy by detecting and trimming potentially distorting outliers in the input data using a robust, likelihood-based approach and by rigorously testing and adjusting for bias covariates that capture systematic variability across known input study characteristics. Additionally, they quantify remaining heterogeneity across input studies that is otherwise not explained, which is incorporated into uncertainty estimates serving as the basis for generating a Burden of Proof Risk Function (BPRF), Risk Outcome Score (ROS), and star ratings – all reflective of both the strength of the association and the strength of the underlying evidence. These measures of evidence strength generated by the Burden of Proof approach provide a crucial complement to existing approaches that evaluate strength of evidence based on expert judgement¹⁷⁶. The methods have already been employed to evaluate health impacts linked with childhood sexual violence and intimate partner violence¹⁰, active¹⁷⁷ and passive smoking¹⁷⁸, chewing tobacco¹⁷⁹, high systolic blood pressure¹⁸⁰, and intake of unprocessed red meat¹⁸¹ and vegetables¹⁸². Our analysis further expands the Burden of Proof literature to include additional forms of violence against children, producing estimates using the same metrics and thus directly comparable to prior Burden of Proof findings, establishing violence against children as a health risk factor akin to smoking and high systolic blood pressure.”

4. This manuscript could be improved through more succinct reporting in all sections. At present the length and amount of information are prohibitive in understanding this work and its implications.

We reviewed the manuscript to ensure that our messages are being communicated as concisely as possible. We made numerous edits for brevity throughout the manuscript; however, we were limited by the breadth of information requested and detail necessary to clearly communicate our methodology and findings.

More minor suggestions are included below:

Abstract

5. The term “violence against children” doesn’t seem to be the best term for the exposure, given this study also included neglect and psychological forms of maltreatment. I would suggest another term that encompasses all the exposures better.

We recognize that there has been and there continues to be active discussion in the field about the use of the term “violence against children” versus the term “childhood maltreatment”. Given the lack of consensus, and the fact that the International Classification for Violence Against Children (ICVAC)^z, has been adopted by UNICEF, WHO, as well as national governments, we

have opted to use the terminology included in ICVAC in our manuscript. This ensures consistency with the most recent UN publications, including the ICVAC, as well as other global efforts such as the recent Global Ministerial Conference on Ending Violence Against Children. While neglect and psychological violence have been considered forms of maltreatment distinct from violence by some experts in the field, the ICVAC explicitly includes these categories within their definitions of violence against children and also notes that “maltreatment” is considered synonymous with “violence” in their conceptualization. Given that regardless of which term we use, there will be negative reactions by some in the field, we have opted to adopt the ICVAC use of terminology as this has been extensively discussed and endorsed by national and global policy makers. We recognize that this is controversial and have added some details clarifying our use of the term in the Methods (lines 496-501):

Lines 496-501: “The ICVAC guidance does not distinguish between violence against children and childhood maltreatment but, rather, suggests that these terms are interchangeable and synonymous. Within this framework, childhood exposure to violence is categorized as any exposure occurring before the age of 18. The present analyses similarly use this demarcation line of 18 years old and adopt the terminology presented in the ICVAC. The ICVAC classifies VAC based on the nature of violent acts, defining each act as either the commission or omission of intentional behavior.”

^z UNICEF. International Classification of Violence against Children (ICVAC). *UNICEF DATA* <https://data.unicef.org/resources/international-classification-of-violence-against-children/> (2024).

Methods

6. It was not clear to me how the star rating was determined. There is a reference to previously published benchmarks but this should be briefly summarised in the main text of this manuscript

We appreciate the request for more details to be presented in the main text of the manuscript. The star rating is determined based on Risk-Outcome Score thresholds that reflect the strength of association and evidence. The star rating is determined based on Risk-Outcome Score thresholds that align with clinically and methodologically relevant minimum excess risk values. As requested, we have added more detail regarding these thresholds in the Methods, and we briefly summarized the published literature^a in which they were established (lines 644-651):

Lines 644-651: “The star ratings aim to aid in the interpretation and comparison of the ROS findings and range from one to five stars, adhering to previously published benchmarks set within the BPRF methodology²⁹. These thresholds were established to align with minimum excess risk values determined in consultation with clinical and methodological experts. For example, a one-star rating indicates weak evidence of association where there is a clear need for further research that may change the assessment of risk as the most conservative interpretation of existing data suggests the possibility of no excess risk. A two-star rating similarly indicates relatively weak evidence of an association with up to 15% increased risk.”

^a Zheng, P. et al. The Burden of Proof studies: assessing the evidence of risk. *Nat Med* **28**, 2038–2044 (2022).

7. It would be helpful to include a table listing the outcomes that were searched for, and found studies for. Also specify what outcomes are included in the term “gynecological diseases”

Thank you for mentioning this. Our systematic review and associated search strings were not outcome-limited, but rather, the search strings used were designed to capture all potential outcomes published related to forms of childhood or gender-based violence. Based on the suggestion by the Reviewer, we have added a new table in the Supplementary Information (Supplementary Table S4) listing all the outcomes that were identified through the systematic review as being linked to exposure to violence, including the outcomes for which we found only 1 or only 2 studies and thus were unable to conduct the Burden of Proof analysis. Regarding gynecological diseases, a full outcome definition, including related outcomes, can now be found in Supplementary Table S4, in addition to which we have included some examples of the component outcomes in the Methods (lines 536-538):

Lines 536-538: "Similarly, herpes and syphilis-related outcome definitions were attributed to the broader category of sexually transmitted infections, and endometriosis and uterine fibroids were included under the category of gynecological diseases following GBD cause groupings."

8. It is also important to acknowledge potential genetic and environmental confounding in the relationship between childhood violence and outcomes e.g. schizophrenia. I direct the authors to literature investigating this issue:.

- Baldwin, J. R., Sallis, H. M., Schoeler, T., Taylor, M. J., Kwong, A. S. F., Tielbeek, J. J., Barkhuizen, W., Warriar, V., Howe, L. D., Danese, A., McCrory, E., Rijdsdijk, F., Larsson, H., Lundström, S., Karlsson, R., Lichtenstein, P., Munafò, M., & Pingault, J. B. (2023). A genetically informed Registered Report on adverse childhood experiences and mental health. *Nat Hum Behav*, 7(2), 269-290. <https://doi.org/10.1038/s41562-022-01482-9>
- Baldwin, J. R., Wang, B., Karwatowska, L., Schoeler, T., Tsaligopoulou, A., Munafò, M. R., & Pingault, J.-B. (2023). Childhood Maltreatment and Mental Health Problems: A Systematic Review and Meta-Analysis of Quasi-Experimental Studies. *American Journal of Psychiatry*, appi.ajp.20220174. <https://doi.org/10.1176/appi.ajp.20220174>

Thank you for this suggestion. We added language about this important point to the Discussion (lines 396-400), including citing the first paper suggested by the reviewer (reference 175). The second paper served as a useful reference for our introduction contextualizing our work (references 9). Both papers highlight an important discussion point that provides further nuance to our results.

Lines 396-400: "another limitation of our study is that we were unable to disaggregate the effects of violence by age of exposure, ethnicity, or other demographic variables due to the constraints in the available data. This is a critical area for future research, as some groups of children may be at an increased risk of violence. Understanding differential impacts, both among high-risk groups and across different epigenetic contexts¹⁷⁵, is essential for tailored interventions."

9. Why were 3 studies chosen as the minimum? The rationale should at least be included

We limited our analysis to only risk-outcome pairs with a minimum of 3 relevant studies because analyses with fewer studies are at greater risk of undue influence from the results of any single study. The 3-study limit is intended to make the analyses less sensitive to any one study and therefore more stable. This decision is applied to all analyses that use the Burden of Proof methodology. We have included this rationale in the Methods (lines 518-519):

Lines 518-519: “the minimum number of data points needed to reasonably evaluate strength of evidence without undue influence by a single study and its design characteristics”

10. It is mentioned that a protocol was published – whether it was prospectively published prior to beginning the systematic review should be included.

We updated our language in the Methods Systematic Review section (lines 464-466) to clarify that the protocol was prospectively published prior to the beginning of the systematic review:

Lines 464-466: “The systematic review from which we drew the inputs for the present investigation was conducted in line with a prospectively published protocol (PROSPERO: CRD42022299831)¹⁸⁵.”

Results

11. Page 7 line 147: suggest including the possible range for ROS to aid interpretation of a score of 0.25

As requested, we have added the ROS range to lines 155-157 (the line numbers cited by the reviewer have changed due to the substantial edits made to the manuscript) to provide context for the values obtained:

Lines 155-157: “Among the 15 analyzed health outcomes, mental health and substance use disorders made up the bulk of the five outcomes with the highest-rated (two stars; ROS between the 0.00 and 0.14 predefined thresholds²⁹) associations with exposure to physical violence in childhood.”

12. Page 7 lines 147-150 seem better suited to the methods section

We already provide detailed descriptions of the key parameters explained in these lines in the Methods section (lines 632-655), and thus, we have kept these lines as a prior in the Results. However, we did reframe the lines suggested by the reviewer (now lines 141-147) to better suit the Results section:

Lines 141-147: “Key analytic parameters, including pooled relative-risk (RR) estimates and surrounding uncertainty—both without between-study heterogeneity/gamma included in uncertainty estimates, aligning with traditional meta-analyses, and with between-study heterogeneity/gamma included, more fully capturing the heterogeneity across underlying studies—are reported for each risk–outcome relationship in Table 2. We also report a Burden of Proof Risk Function (BPRF) value and corresponding Risk-Outcome Score (ROS) and star rating for each pair.”

Discussion

13. The findings of this study contrast against a wealth of literature that report strong associations between childhood violence and many of the health outcomes reviewed. Therefore, the discussion should include greater attention to why the results of this review were so different.

We substantially revised the language in our methods and discussion to more clearly elucidate the methodological differences and strengths of our meta-analytic approach compared to a conventional meta-analysis (lines 432-448):

Lines 432-448: “This methodology utilizes a meta-regression tool—Bayesian, regularized, trimmed (MR-BRT)—to derive pooled relative risk estimates and associated uncertainty. These methods improve estimation accuracy by detecting and trimming potentially distorting outliers in the input data using a robust, likelihood-based approach and by rigorously testing and adjusting for bias covariates that capture systematic variability across known input study characteristics. Additionally, they quantify remaining heterogeneity across input studies that is otherwise not explained, which is incorporated into uncertainty estimates serving as the basis for generating a Burden of Proof Risk Function (BPRF), Risk Outcome Score (ROS), and star ratings – all reflective of both the strength of the association and the strength of the underlying evidence. These measures of evidence strength generated by the Burden of Proof approach provide a crucial complement to existing approaches that evaluate strength of evidence based on expert judgement¹⁷⁶. The methods have already been employed to evaluate health impacts linked with childhood sexual violence and intimate partner violence¹⁰, active¹⁷⁷ and passive smoking¹⁷⁸, chewing tobacco¹⁷⁹, high systolic blood pressure¹⁸⁰, and intake of unprocessed red meat¹⁸¹ and vegetables¹⁸². Our analysis further expands the Burden of Proof literature to include additional forms of violence against children, producing estimates using the same metrics and thus directly comparable to prior Burden of Proof findings, establishing violence against children as a health risk factor akin to smoking and high systolic blood pressure.”

We also strived to better contextualize our findings within existing literature and highlighted that our overall results are largely consistent with other reports even as we find and account for gaps and heterogeneity in existing literature (lines 285-308):

Lines 285-308: “Particularly given our conservative interpretation of the evidence, the adverse health consequences found by our analysis in association with violence against children—which had already been underscored by some traditional meta-analytical efforts^{141,142}—reinforces the need for public health approaches to encourage the prevention of all forms of childhood violence, to better address the health needs of survivors, and to bolster the evidence base relating health consequences to childhood violence exposure.

According to our results, one area of considerable increased risk associated with VAC is mental disorders. Consistent with the literature, we found that all forms of violence were associated with significantly greater risks for major depressive disorder and anxiety disorders^{9,143,144}. [...]

Recent meta-analytical efforts have also showcased substance use disorders as another important category of increased health risks, with age-, gender- and violence type-dependent associations to childhood violence exposure^{17,147,148}. When accounting for known and unknown sources of between-study variation and potential bias, we found the risk of alcohol use disorders escalating by a minimum of 45% and 4% following childhood experiences of sexual violence¹⁰ and physical violence respectively.”

14. Table 1 – I’m not sure why policy implications are included as a table and not in text in the discussion.

We have modified our language in the Discussion to more explicitly refer to the policy implications of our findings (lines 272-290):

Lines 272-290: “Using the Burden of Proof analytic framework, we generated intentionally conservative assessments of the associations between exposure to these forms of violence and negative health outcomes based on methods that accounted for both known variability across

study-design characteristics and remaining heterogeneity among input-level effect estimates, yielding highly robust and credible findings. In relation to childhood physical violence, our analysis found significant increased risks for 11 of 15 health outcomes analyzed for individuals exposed to this form of violence. Associations between physical violence and major depressive disorders, IHD, substance use disorders, and eating disorders received two-star (moderately weak) ratings, a measure that summarizes both the strength of the association and strength of the underlying evidence. Similarly, we found significant associations between childhood psychological violence and 10 of 11 health outcomes analyzed, with the associations with drug use disorders, migraine, and gynecological issues also receiving two-star ratings. We also identified significant associations between childhood neglect and 6 of 9 health outcomes analyzed, with the relationship with anxiety being rated two stars. Particularly given our conservative interpretation of the evidence, the adverse health consequences found by our analysis in association with violence against children—which had already been underscored by some traditional meta-analytical efforts^{141,142}—reinforces the need for public health approaches to encourage the prevention of all forms of childhood violence, to better address the health needs of survivors, and to bolster the evidence base relating health consequences to childhood violence exposure.”

However, we have retained Table 1 with the policy implications in alignment with the Burden of Proof body of work and at the request of the Nature family journals. We are happy to modify the role of this table if the editor of this piece would prefer. For now, this table provides a brief and easy-to-identify summary of the key findings and take-aways of the paper beyond the abstract, which are further elaborated throughout the Discussion.

Reviewer #3:

Thank you for the opportunity to review this paper. It presents a novel analytical framework to summarise the evidence of associations between violence against children (aka maltreatment) and health outcomes. The paper employs an interesting analytical approach building on previously published work already including child sexual abuse. I have some comments, which I hope may be helpful in revising the manuscript.

Thank you to the reviewer for their supportive assessment of our novel analytical approach to the crucial topic of violence against children. We have incorporated the reviewer’s comments into our manuscript and appreciate the opportunity to strengthen the reporting of our analysis.

1. There is extensive evidence synthesis work on the links between violence against children (aka childhood maltreatment) and various health outcomes. The literature is particularly saturated for the associations with mental health outcomes, which is extensively discussed here, too. The authors should clarify the novelty of their work in the context of the literature, which they consider to a minimal extent. In particular, I appreciate they use a novel metric, but I am unclear whether this offers advantages over previous work (please see other comments below).

We appreciate the reviewer highlighting this point of ambiguity. Throughout the updated manuscript, including in the Introduction (lines 109-115), Discussion (lines 272-276; 285-290), and Methods (lines 432-448), we provided a more detailed and explicit overview of the strengths of the Burden of Proof methodology over approaches used in existing meta-analyses:

Lines 109-115: “This novel analytical framework incorporates—among other systematic modeling components—covariate selection and adjustment to account for known variation in

input study characteristics and further quantifies remaining ‘unexplained’ between-study heterogeneity using random effects modeling (e.g., gamma and its uncertainty). Therefore, this method more fully accounts for heterogeneity compared to traditional meta-analytic approaches, providing uniquely robust, credible, and well-specified evidence for policymakers and public health professionals dedicated to reducing VAC and its impacts.”

Lines 272-276; 285-290: “Using the Burden of Proof analytic framework, we generated intentionally conservative assessments of the associations between exposure to these forms of violence and negative health outcomes based on methods that accounted for both known variability across study-design characteristics and remaining heterogeneity among input-level effect estimates, yielding highly robust and credible findings. [...] Particularly given our conservative interpretation of the evidence, the adverse health consequences found by our analysis in association with violence against children—which had already been underscored by some traditional meta-analytical efforts^{141,142}—reinforces the need for public health approaches to encourage the prevention of all forms of childhood violence, to better address the health needs of survivors, and to bolster the evidence base relating health consequences to childhood violence exposure.”

Lines 432-448: “This methodology utilizes a meta-regression tool—Bayesian, regularized, trimmed (MR-BRT)—to derive pooled relative risk estimates and associated uncertainty. These methods improve estimation accuracy by detecting and trimming potentially distorting outliers in the input data using a robust, likelihood-based approach and by rigorously testing and adjusting for bias covariates that capture systematic variability across known input study characteristics. Additionally, they quantify remaining heterogeneity across input studies that is otherwise not explained, which is incorporated into uncertainty estimates serving as the basis for generating a Burden of Proof Risk Function (BPRF), Risk Outcome Score (ROS), and star ratings – all reflective of both the strength of the association and the strength of the underlying evidence. These measures of evidence strength generated by the Burden of Proof approach provide a crucial complement to existing approaches that evaluate strength of evidence based on expert judgement¹⁷⁶. The methods have already been employed to evaluate health impacts linked with childhood sexual violence and intimate partner violence¹⁰, active¹⁷⁷ and passive smoking¹⁷⁸, chewing tobacco¹⁷⁹, high systolic blood pressure¹⁸⁰, and intake of unprocessed red meat¹⁸¹ and vegetables¹⁸². Our analysis further expands the Burden of Proof literature to include additional forms of violence against children, producing estimates using the same metrics and thus directly comparable to prior Burden of Proof findings, establishing violence against children as a health risk factor akin to smoking and high systolic blood pressure.”

This approach produces several novel metrics that uniquely allow us to 1) systematically evaluate evidence across risk-outcome pairs, 2) quantify and incorporate measures of consistency in the literature into our assessment of the associations, and 3) account for known and unknown sources of potential biases and 4) apply the same methodology to exposure to violence as used in all other risk factors in the Global Burden of Disease study, thus enabling us to propose inclusion of this very important risk factor in the GBD study. Furthermore, while there are previous systematic reviews, we have not found any that considered all health outcomes associated with exposure to violence during childhood at the same time and using the same methods. We also wanted to mention that during the revision stage, we updated our review to span through the beginning of 2024, making it a very updated synthesis of existing evidence compared to other previously published works.

2. The systematic review methodology is overall solid. However, I note that the analyses do not seem to include important studies in the area (e.g., Dunedin Study, E-Risk Study), which raises concerns about the comprehensiveness of the data included in the analyses.

Our prospectively published protocol⁹ was formally published and adhered to throughout the systematic review process, which allowed us to identify all relevant studies that met our inclusion and exclusion criteria (described in the Methods – line 472-478) pertaining to health outcomes associated with gender-based violence and violence against children. In response to the Reviewer’s comment, we re-reviewed all identified publications that used data from the Dunedin or E-Risk Cohorts and the reasons for exclusion have been re-examined and confirmed. It is worth noting that our study uses the diseases and conditions included in the Global Burden of Disease study, and so the outcomes that we analyze need to be conditions/diseases that are in the cause hierarchy of the GBD study.

Specifically, among the Dunedin cohort studies, there were six unique studies which utilized data from the Dunedin Cohort and seven unique studies utilizing the E-Risk Cohort data. Of these studies, none met all of our inclusion criteria, and thus were excluded from our analyses. Of the six studies utilizing Dunedin Cohort data, three did not include outcomes that were relevant to our analysis but rather focused on outcomes outside of the scope of the present analysis, like driving behaviors and inflammation; one did not provide effect sizes and related uncertainties or underlying data, which is essential for inclusion; one did not provide data on any of our specific exposures of interest; and we were unable to access the full text of the last publication through any of our resources. Of the seven studies utilizing the E-Risk Cohort data, four focused only on outcomes that are not outcomes that are included in our analysis, and three did not provide data on any of our exposures of interest. The exclusion reasons and references are provided in more detail below.

Dunedin Study	
Reference captured in our systematic review	Exclusion reason
Bourassa, K. J. et al. Lower Cardiovascular Reactivity Is Associated With More Childhood Adversity and Poorer Midlife Health: Replicated Findings From the Dunedin and MIDUS Cohorts. Clinical Psychological Science 9 , 961–978 (2021).	Missing essential data necessary for analysis, including effect sizes and related uncertainties
Baldwin, J. R. et al. Population vs Individual Prediction of Poor Health From Results of Adverse Childhood Experiences Screening. JAMA Pediatrics 175 , 385–393 (2021).	Exposure did not meet our inclusion criteria: Defined as cumulative ACE scores
Gulliver, P. & Begg, D. Personality factors as predictors of persistent risky driving behavior and crash involvement among young adults. Injury Prevention 13 , 376–381 (2007).	Outcome did not meet our inclusion criteria: Driving behavior and crash involvement
Danese, A. Biological embedding of child stress through inflammation. Journal of	Outcome did not meet our inclusion criteria: Biological stress levels evaluated as inflammation

Psychosomatic Research 74 , 543–544 (2013).	
Danese, A. S.15.02 Childhood maltreatment and the course of depression: the role of inflammation. European Neuropsychopharmacology 23 , S135 (2013).	Unable to retrieve full text for review
Davidson, R. J. Childhood Adversity and the Brain: Harnessing the Power of Neuroplasticity. Biological Psychiatry 90 , 143–144 (2021).	Outcome did not meet our inclusion criteria: Structural neurological changes
E-Risk Cohort	
Reference captured in our systematic review	Exclusion reason
Matthews, T. et al. A longitudinal twin study of victimization and loneliness from childhood to young adulthood. Development and Psychopathology 34 , 367–377 (2022).	Outcome did not meet our inclusion criteria: Loneliness
Schaefer, J. D. et al. Adolescent Victimization and Early-Adult Psychopathology: Approaching Causal Inference Using a Longitudinal Twin Study to Rule Out Noncausal Explanations. Clinical Psychological Science 6 , 352–371 (2018).	Outcome did not meet our inclusion criteria: Internalized disorders and thought disorders that did not meet our diagnostic criteria
Baldwin, J. R. et al. Population vs Individual Prediction of Poor Health From Results of Adverse Childhood Experiences Screening. JAMA Pediatrics 175 , 385–393 (2021).	Exposure did not meet our inclusion criteria: Defined as cumulative ACE scores
Baldwin, J., Arseneault, L. & Danese, A. Childhood bullying and adiposity in young adulthood: Findings from the E-Risk Longitudinal Twin Study. Psychoneuroendocrinology 61 , 16 (2015).	Exposure did not meet our inclusion criteria: Bullying
Baldwin, J., Arseneault, L. & Danese, A. Childhood violence victimisation predicts elevated inflammation levels in young women independent of latent genetic influences. Psychoneuroendocrinology 71 , 18 (2016).	Outcome did not meet our inclusion criteria: Inflammation levels
Baldwin, J. R., Arseneault, L. & Danese, A. Childhood victimization and inflammation in young adulthood: A genetically sensitive cohort study. Psychoneuroendocrinology 83 , 61 (2017).	Outcome did not meet our inclusion criteria: Inflammation levels

Baldwin, J. R., Arseneault, L. & Danese, A. 4.38 Adolescent Victimization and Self-Injurious Thoughts and Behaviors: A Genetically Sensitive Cohort Study. Journal of the American Academy of Child & Adolescent Psychiatry 57, S216–S217 (2018).	Exposure did not meet our inclusion criteria: Bullying
--	---

We believe that our systematic review and resulting analyses are comprehensive and complete following our established inclusion and exclusion criteria.

^y Spencer, C. N. *et al.* Estimating the global health impact of gender-based violence and violence against children: a systematic review and meta-analysis protocol. *BMJ Open* **12**, e061248 (2022).

3. Authors should clarify if there were any studies omitted for lack of information in the original papers / after requesting information from the authors.

Thank you for this suggestion. We have added language in the Results on the studies excluded (lines 122-124). This includes 438 studies excluded due to the absence of necessary information in the original papers:

Lines 122-124: “All other studies were found to be ineligible, including 438 that failed to report necessary data (Supplementary Figure S1).”

The full breakdown of exclusionary criteria and number of studies excluded for each criterion be found in the PRISMA diagram provided (Supplementary Figure S1). We did not solicit non-published information from the authors of underlying papers, but we hope to supplement this work in the future. However, these data fell outside of the scope of the present analysis, which focused on studies in the peer-reviewed literature.

4. On page 24, can the authors explain how they ensure that 'the most adjusted effect size' was always the best choice for data selection? For example, would 'the most adjusted effect size' be the one that adjusts for the co-occurrence of other maltreatment types? Or genetic influences on the health outcomes considered? These are important choices with considerable implications for the interpretation of the results.

We added further clarification in the Methods (lines 541-566) regarding our data point selection. The most adjusted effect size was defined as the effect size that controlled for the greatest number of potentially confounding variables, characterized as the “fully adjusted” model in most studies. Depending on the study in question, this may include the co-occurrence of other violence types or genetic characteristics, depending on the outcome in question and study design. However, when there are multiple adjusted effect sizes presented in a single study, we developed a step-wise approach that prioritized data points that accounted for the co-occurrence of other violence types, as well as other exposure, outcome, and demographic characteristics. These are described in detail:

Lines 541-566: “From each study, the effect sizes that adjusted for the greatest number of relevant potential confounding variables associated with a given violence type and health outcome combination were selected to form our input datasets. When multiple adjusted effect

sizes were available, we implemented a further sequential data point selection process to identify those with the closest exposure and outcome definitions to our reference definitions, the least restrictive perpetrator type, and the broadest sample. First, to reduce the potential impact of co-occurrent forms of violence, we prioritized analytical samples where the exposed groups were limited to individuals who had only experienced the violence type of interest. If information regarding restrictions on the case definitions was not available or no restriction was made on the basis of other exposures to violence, we selected effect sizes that were controlled for alternative exposures to violence over the effect sizes from the same study from samples that were not explicitly restricted or adjusted for alternative exposures. For studies with multiple recall periods associated with an exposure, such as one effect size for exposure to violence in the past year and another for exposure to violence at any time during childhood, we selected the observations that corresponded to the longest recall period. Furthermore, we selected the observations for outcomes that most closely matched the GBD cause definitions if a study reported on effect sizes for different outcome sub-types or groupings.

For studies that reported on perpetrator-specific violence and on general violence perpetrated by anyone or an aggregated perpetrator grouping, we selected the observations for the broadest perpetrator type for our primary analysis. We conducted sensitivity analyses for violence perpetrated by family/household members when sufficient perpetrator-specific data was available. For studies that reported on other types of sub-group analyses in addition to a primary analysis, such as study site-specific analyses, we prioritized observations from the overall sample. Lastly, we selected observations that were derived from samples of both male and female participants over those that were sex-stratified when both were available from the same study. We conducted sex-specific sensitivity analyses selecting the sex-stratified observations when feasible to examine differences in risk and available evidence by sex.”

When possible, we conducted sensitivity analyses evaluating the impact of these decisions and reported these findings throughout the Results section and the appendix. Furthermore, the potential impact of co-occurrence and epigenetic factors are explored in the Limitations (lines 382-394; 396-400) for studies that did not control for these factors:

Lines 382-394: “Additionally, our analysis focused on reports of distinct forms of child abuse, and it did not account for the co-occurrence of violence types that often take place together^{172,173}. Co-occurrence of different forms of violence may act as compounding risk factors¹⁷⁴, particularly when considering the outcomes we found to be shared across physical and psychological violence and neglect, like major depressive disorder. Unfortunately, we found that most studies did not explicitly state whether their case definitions were exclusive to the type of violence in question, if they also included children who had concurrently experienced other forms of violence, or if they had even investigated the possibility of poly-victimization. On occasion, a study provided multiple data points accounting for different forms of violence. In these cases, we selected the data points that were explicitly restricted to individuals who had only experienced the violence type in question or that controlled for exposure to another form of violence. However, there was insufficient data on specific combinations of violence to fully explore the consequences of co-occurrence.”

Lines 396-400: “Third, another limitation of our study is that we were unable to disaggregate the effects of violence by age of exposure, ethnicity, or other demographic variables due to the constraints in the available data. This is a critical area for future research, as some groups of children may be at an increased risk of violence. Understanding differential impacts, both

among high-risk groups and across different epigenetic contexts¹⁷⁵, is essential for tailored interventions.”

5. The authors did not seem to include any information on the source of maltreatment, which has proven to be an important characteristic in modifying associations with health outcomes. For example, prospective and retrospective measures identify different people (<https://jamanetwork.com/journals/jamapsychiatry/fullarticle/2728182>) and are differentially associated with psychopathology (<https://jamanetwork.com/journals/jamapsychiatry/article-abstract/2818046>). As such, ignoring these differences may lead to measurement error, which would dilute the associations found. It is also a critical point for interpretation of the findings: while prevention of the experiences is necessary, this is unlikely in itself to address the burden of illness discussed (which would likely additionally require addressing the individual experiences).

Thank you for raising this important point. We enumerated full details regarding exposure ascertainment and definitions for each of our included studies in the Supplementary Information (Supplemental Table S3). When possible, we generated bias covariates capturing variation in case definitions and study designs and tested the covariates for potential inclusion in the final models (more details are provided in the Methods lines 582-608). In response to the reviewer’s comment, we have elaborated further on various measurement challenges, including prospective and retrospective measures, in the Discussion section (lines 354-368) to provide more context as to how these issues may affect the interpretation of our results:

Lines 354-368: “VAC is a multifaceted issue that can be characterized by the nature of the violent act, the impact on the victim, the relationship between the victim and the perpetrator(s), and the context in which the violence occurs. There are also substantial challenges when measuring these characteristics, including potential recall bias or dissociative amnesia when self-reporting exposure and the under-reporting of violence through formal channels, which can affect the magnitude of the associations identified^{161,162}. To address the challenge of synthesizing heterogenous data and enhance the comparability of our findings with future research, we operationalized the nature of violence in accordance with the ICVAC³⁰. Additionally, to mitigate the influence of differing definitions related to specific categories of perpetrators, we relied on bias covariates. However, our ability to control for other dimensions of violence exposure and assessment was constrained by the dearth of data and, in some cases, by the absence of a clear gold-standard practice for exposure assessment. For example, both prospective (in the form of formal reports) and retrospective assessments of childhood violence have substantial weaknesses that may result in under-reporting or skewed reporting. Consequently, this limitation may have contributed to the unexplained between-study heterogeneity estimated as part of the Burden of Proof methodology.”

The reviewer’s comment highlights the importance of addressing the needs of survivors, in addition to preventing violence and we have further elaborated on this in our Discussion (lines 329-335):

Lines 329-335: “This gap in available evidence highlights an urgent need to invest in well-designed studies from a diverse set of countries and contexts that can elucidate the full spectrum of VAC’s health implications and how survivors may experience the consequences of violence long after their experiences. As highlighted by the World Health Organization (WHO) action plan to strengthen health systems’ capacity to address abuse against children¹⁵³, timely

and compelling evidence detailing the burden of VAC is critical for tailoring effective prevention and response strategies.”

6. In the second paragraph on page 26 (section on 'Testing and adjusting for biases across study designs and characteristics'), the algorithm / statistical approach needs to be clarified for the readers.

We added more details regarding the Lasso technique used to test for bias covariates to be included (lines 597-602):

Lines 597-602: “Using Burden of Proof methods, we systematically tested the bias covariates utilizing an automated selection algorithm that employs a step-by-step technique to identify covariates that significantly affect risk estimates. In brief, potential bias covariates are sequentially ranked using a Lasso approach^{188,189}, and then added one-by-one – starting with the highest ranked – as interaction terms with the crude pooled relative risk to the linear meta-regression model. This process is terminated as soon as a bias covariate is added that is not significantly associated with the effect size.”

Our clarification aims to elucidate the technique but does not delve into the mathematical equations that underlie the methods, which are fully delineated in the Burden of Proof methods paper^a.

^a Zheng, P. et al. The Burden of Proof studies: assessing the evidence of risk. *Nat Med* **28**, 2038–2044 (2022).

7. Authors should provide a clearer and more accessible description of their methods involving the proof of risk function and model validation for non-specialist (pages 27-28), possibly referring to further supplementary materials. This text presents the innovation in the paper, and very few readers will be able to understand it and evaluate it. Beyond the steps taken, authors should provide clearer rationale for each of the step so as to walk the readers through their work.

Thank you for this suggestion. We have substantially revised the language used in the pages pointed out by the reviewer (lines 633-653) to more clearly communicate our methodology to a non-statistical audience:

Lines 633-653: “The BPRF reflects the most conservative estimate of association between each of the exposures and the health outcome of interest that is consistent with the available evidence, and can be compared across different risk-outcome pairs. For harmful risk factors such as childhood physical abuse, psychological abuse, and neglect, the BPRF is calculated as the 5th quantile of draws closest to the null from the distribution defined by the relative risk UIs inclusive of between-study heterogeneity. From the BPRF, we derived the Risk-Outcome Score (ROS) as the signed natural log(BPRF) divided by two and a conservative estimate of the minimum increased risk of the health outcome due to exposure to the risk factor. The ROS reflects both the magnitude of the risk-outcome association and the consistency of findings studies informing the association. A large positive ROS nearing 1 indicates a strong association supported by consistent evidence, whereas a negative or small (< 0.14) positive ROS reflects a weak association and/or a lack of consistent evidence of an association. ROS values can also be translated into star rating categories. The star ratings aim to aid in the interpretation and comparison of the ROS findings and range from one to five stars, adhering to previously

published benchmarks set within the BPRF methodology²⁹. These thresholds were established to align with minimum excess risk values determined in consultation with clinical and methodological experts. For example, a one-star rating indicates weak evidence of association where there is a clear need for further research that may change the assessment of risk as the most conservative interpretation of existing data suggests the possibility of no excess risk. A two-star rating similarly indicates relatively weak evidence of an association with up to 15% increased risk. Additional stars reflect increasing evidence of an association up to a five-star rating, which suggests very strong evidence and a conservative estimate of an increase in risk by at least 85%.”

We have also added lines in every step of the Methodology (lines 541-556; 572-574; 607-608; 610-611; 621-624; 633-635) to explicitly state the rationale for each and the added benefit. We hope our modifications will enhance readers’ understanding of this methodology and its strengths relative to a traditional meta-analytic approach.

8. The estimates presented look comparatively small and weak beyond a few examples. In contrast, the authors conclude that policymakers and health professionals should prioritise the elimination of all forms of violence against children (which, of course, should be a priority even at a moral level and has been highlighted many times previously). It would be helpful if authors could better link their results and conclusions.

We appreciate this suggested improvement to the Discussion and have made edits to more directly link the implications of our findings to our conclusions (lines 285-290; 317-335):

Lines 285-290: “Particularly given our conservative interpretation of the evidence, the adverse health consequences found by our analysis in association with violence against children—which had already been underscored by some traditional meta-analytical efforts^{141,142}—reinforces the need for public health approaches to encourage the prevention of all forms of childhood violence, to better address the health needs of survivors, and to bolster the evidence base relating health consequences to childhood violence exposure.”

Lines 317-335: “Although our Burden of Proof findings provide highly credible confirmation of the adverse health consequences associated with each form of childhood violence and despite the pooled effect sizes as large as 3.08, most of the significant risk–outcome relationships identified in our study received moderately weak or weak star ratings, mirroring the findings for CSA¹⁰. This is primarily due to the weak and/or inconsistent evidence base, i.e., small numbers of input studies and high degrees of between-study heterogeneity. Out of the 35 risk–outcome pairs we assessed, almost half of our analyses were based on five or fewer studies, and a considerable portion only met the minimal threshold of three studies required for evaluation. The paucity of research, particularly on the psychological violence and neglect dimensions of VAC, poses challenges for comparing the negative impacts of different forms of violence across the same health outcomes. Anxiety disorders, major depressive disorders, alcohol use disorders, drug use disorders, schizophrenia, type 2 diabetes mellitus, asthma, and self-harm were the only outcomes with sufficient data to analyze associations with all categories of childhood violence, including sexual¹⁰, physical, and psychological violence, and neglect. This gap in available evidence highlights an urgent need to invest in well-designed studies from a diverse set of countries and contexts that can elucidate the full spectrum of VAC’s health implications and how survivors may experience the consequences of violence long after their experiences. As highlighted by the World Health Organization (WHO) action plan to strengthen health

systems' capacity to address abuse against children¹⁵³, timely and compelling evidence detailing the burden of VAC is critical for tailoring effective prevention and response strategies.”

One important tenant in our analytical approach is to take into account the amount of available evidence, the variation across studies, as well as the average effect size found in the studies. For some of the outcomes we were able to explore, when looking at all three components at the same time, the strength of the evidence is classified as weak by the Burden of Proof methodology. As more studies get published for these associations that are currently assigned a one-star or a two-star rating, the average effect size and the star rating might change and we intend to keep updating this systematic review annually. As the Reviewer also highlights, the small effect size and/or the sparsity of studies for some of the associations included, does not in any way take away from the moral imperative to prevent violence against children. We have rephrased the Discussion around the implications of our findings and we hope that it now more accurately reflects the interpretation of our results.

9. The authors looked at associations with different types of childhood maltreatment but used a different label here (violence against children). I don't think that the label works because the experience includes neglect, which is not by definition a form of violence. I appreciate this may be a terminology suggested by UN, but it will not be familiar to most clinicians and public health practitioners. The use of different terms may also add confusion to an already fragmented literature.

Our team recognizes that there is a lot of active discussion, debate and lack of consensus in the field about using the term “violence against children” versus the term “childhood maltreatment,” particularly given the point highlighted here regarding the nuances of neglect as a category of harm. We have opted to use the term “violence against children” to reflect the terminology used in the most recent UN publications, including the International Classification for Violence Against Children (ICVAC)^z and the recent Global Ministerial Conference on Ending Violence Against Children. While the reviewer is correct that neglect has been considered by many a form of maltreatment distinct from violence, the UN explicitly includes these categories within their definitions of violence against children, while the ICVAC notes that “maltreatment” is considered synonymous with “violence” in their conceptualization. As a result, we feel like the term “violence against children” accurately encapsulates the diverse forms of harm captured in our analyses and reflects the current language being used by the field. However, we take the reviewer’s point that the use of terminology unfamiliar to some may generate confusion, therefore we have added language in the Methods clarifying our use of terminology (lines 496-501):

Lines 496-501: “The ICVAC guidance does not distinguish between violence against children and childhood maltreatment but, rather, suggests that these terms are interchangeable and synonymous. Within this framework, childhood exposure to violence is categorized as any exposure occurring before the age of 18. The present analyses similarly use this demarcation line of 18 years old and adopt the terminology presented in the ICVAC. The ICVAC classifies VAC based on the nature of violent acts, defining each act as either the commission or omission of intentional behavior.”

^z UNICEF. International Classification of Violence against Children (ICVAC). *UNICEF DATA* <https://data.unicef.org/resources/international-classification-of-violence-against-children/> (2024).

Response to Reviewers for NATHUMBEHAV-24020706:

Health effects associated with exposure of children to physical violence, psychological violence, and neglect: a Burden of Proof study

REVIEWER COMMENTS:

Reviewer #1:

I believe the authors have been thoroughly responsive to reviewers comments. I appreciate their edits to increase accessibility to the unique add of their analytical approach to the literature. I have no additional comments for further revision.

We appreciate Reviewer #1's kind words and feedback throughout the peer-review process.

Reviewer #2:

The reviewers have greatly strengthened the manuscript by highlighting the advantages of the burden of proof methodology. They have satisfactorily addressed most of my comments, yet two minor points remain.

Thank you to Reviewer #2 for their assistance in improving our manuscript to its current version and for their acknowledgment of the improvements made. We are pleased to have addressed the major comments presented by this reviewer to their satisfaction and hope to similarly provide suitable responses to the two pending minor comments.

I and other reviewers commented on the use of the term "violence against children" when it includes exposures like neglect. I respect that the authors wish to retain the term "violence against children", but their response doesn't adequately address the issue of describing neglect as an act of violence. The paragraph they have included in response should include mention of neglect specifically, perhaps at the end of the second sentence, or the final sentence.

Lines 496-501: "The ICVAC guidance does not distinguish between violence against children and childhood maltreatment but, rather, suggests that these terms are interchangeable and synonymous. Within this framework, childhood exposure to violence is categorized as any exposure occurring before the age of 18. The present analyses similarly use this demarcation line of 18 years old and adopt the terminology presented in the ICVAC. The ICVAC classifies VAC based on the nature of violent acts, defining each act as either the commission or omission of intentional behavior."

We have modified the section suggested by the reviewer to clarify the inclusion of neglect in the ICVAC guidance and, consequently, in our use of the term "violence against children." This section now reads as follows:

Lines 518-524: *"The ICVAC guidance does not distinguish between violence against children and childhood maltreatment but, rather, suggests that these terms are interchangeable and synonymous. In doing so, it considers both forms of maltreatment, like neglect, and forms of direct abuse of children as violence. Within this framework, childhood exposure to violence is categorized as any exposure occurring before the age of 18. The present analyses similarly use*

this demarcation line of 18 years old and adopt the terminology presented in the ICVAC. The ICVAC classifies VAC based on the nature of violent acts, defining each act as either the commission or omission of intentional behavior (including neglect)."

Additionally, we have included this clarification in our Introduction, more clearly stating that the term "Violence Against Children", in our study, encompasses "*forms of direct abuse or neglect of children*" (Line 75).

The second point is to do with acknowledging the potential for genetic and environmental confounding that I raised previously (Reviewer 2, point 8). The authors perhaps misunderstood my point, which was that children with a parent suffering a mental illness or substance use disorder may be more likely to be the victim of violence, but also more likely to develop a mental disorder or other outcome studied due to genetic liability. Therefore, without rigorous assessment, we cannot rule out these potential sources of confounding and say that violence accounts for all of this association when it may be that this is explained in part by the genetic component. So my point was more about mentioning in the discussion that these associations do not reflect causality and controlling for important genetic and environmental confounds would likely temper the magnitude of these associations. I still think this is important to mention briefly in the discussion.

We truly appreciate the reviewer clarifying their point on genetic confounding. The reviewer correctly identified that we had misunderstood their original comment, but their point is well-taken. As the reviewer notes here, we cannot rule out these confounding factors with the data that is available for this meta-analysis, and we have clarified the language addressing this important limitation in the discussion:

Lines 419-424: "Moreover, by not being able to account for different epigenetic contexts, we must consider the possible confounding role of genetics, which may increase familial risk of mental or substance use disorders, like alcohol use disorder, and in turn may increase the risk of violence exposure¹⁷⁶⁻¹⁷⁸. In lieu of more detailed data that accounts for genetic factors, these associations may exacerbate some of our observed effects between VAC and conditions with genetic components."

Otherwise I am satisfied with the authors' response.

Reviewer #3:

In the revised manuscript, the authors failed to explain convincingly the novelty of the statistical methods and the importance of the findings in a language that is accessible to an audience of my peers. Furthermore, despite the statistical advances proposed, the methods still cannot address key concerns in the field including informant effects or genetic confounding. As such, the overall innovation appears limited in the context of the existing literature.

To expand on one of these aspects, in response to my comments as well as comments by other reviewers, the author did not to address the point that, by their nature, the victimisation experiences that they assess as separate are very often interdependent. The authors now explain that the selected, where possible, samples with exposure to individual maltreatment

types (which are not representative of victimisation occurrence in the population); alternatively, they selected effect sizes in analyses adjusting for co-occurrence (which artificially reduces the strength of associations observed modelling poly-victimisation given the natural co-occurrence). As such, this choice is misguided and offers a very partial overview of the field. It also likely contributes explaining why results somewhat differ from the findings of other published in other systematic work.

The reviewer's concerns are noted; however, we differ in the belief that our aim to investigate the distinct health consequences of different forms of violence against children is misguided. As the reviewer rightly notes, there is a close relationship between being exposed to one form of violence and experiencing another, and these co-exposures likely compound the associated health risks associated with any one type of exposure. However, our analysis highlights that even exposure to one specific sub-type of violence against children results in life-long health harms, which is crucial when considering that poly-victimization may not be identified immediately when instances of one form of violence are first documented. Our findings are additionally consistent with other meta-analyses in finding that differences in the long-term health consequences between different forms of violence against children, although the breadth of health outcomes captured in our analysis is broader.¹ As a result, we can firmly conclude that survivors may have different long-term needs depending on the form(s) of their exposure. We have added some textual clarification in our discussion to make it clear that we consider our estimates to be the lower bound of health risk associated with violence against children.

Lines 405-411: "In these cases, we selected the data points that were explicitly restricted to individuals who had only experienced the violence type in question or that controlled for exposure to another form of violence in order to narrow in on the distinct health toll of different types of VAC. While there was insufficient data on specific combinations of violence to fully explore the consequences of co-occurrence, survivors often experience multiple forms of VAC, and these health risks may compound over multiple exposures. As a result, our approach presents only the lowest bound of associated health risk a survivor may experience in the context of poly-victimization."

Furthermore, the reviewer specifically highlights the manner in which we selected data points in cases where a single study reported many eligible effect sizes as problematic. However, we respectfully argue that our approach is consistent with the aim of our analysis—to investigate the distinct health consequences of different forms of violence—and that it does not artificially skew our results. Firstly, the overall selection process applies only to the studies with multiple data points from the same sample, which represents a minority of the studies identified in our systematic review. Selection based on exposure type or adjustment for co-occurrence specifically applies to a very small portion of the studies included because it occurs only after other sources of variability have been accounted for and because studies rarely reported multiple effect sizes that differ in these specific ways. As a result, we do not believe that changing this approach would substantially affect our results. Second, in the rare cases where selection based on exposure type or adjustment applied, we made the determination to select observations that minimized the potential impact of another form of violence to be aligned with our central aim.

¹ Gardner, M. J., Thomas, H. J. & Erskine, H. E. The association between five forms of child maltreatment and depressive and anxiety disorders: A systematic review and meta-analysis. *Child Abuse & Neglect* 96, 104082 (2019).

If the authors still want to use primarily the term 'violence against children', they should also include the term child maltreatment as key term and ideally in the abstract to ensure that the study is index more broadly.

We thank the reviewer for this suggestion. We have added the term child maltreatment to the abstract given Reviewer #3's point on indexing to ensure that the study is visible to the individuals interested in this important topic. We also plan on using the term child maltreatment as a key term.